

**Variations and sources of volatile organic compounds (VOCs)**
**in urban region: insights from measurements on a tall tower**
Xiao-Bing Li[1,2], Bin Yuan[1,2,*], Sihang Wang[1,2], Chunlin Wang[3,4], Jing Lan[3,4], Zhijie
Liu[1,2], Yongxin Song[1,2], Xianjun He[1,2], Yibo Huangfu[1,2], Chenglei Pei[5,6,7,8], Peng
Cheng[9], Suxia Yang[1,2], Jipeng Qi[1,2], Caihong Wu[1,2], Shan Huang[1,2], Yingchang You[1,2],
Ming Chang[1,2], Huadan Zheng[10], Wenda Yang[9], Xuemei Wang[1,2], and Min Shao[1,2]
[1] Institute for Environmental and Climate Research, Jinan University, Guangzhou
511443, China
[2] Guangdong-Hongkong-Macau Joint Laboratory of Collaborative Innovation for
Environmental Quality, Guangzhou 511443, China
[3] Guangzhou Climate and Agrometeorology Center, Guangzhou, 511430, China
[4] Southern Marine Science and Engineering Guangdong Laboratory (Zhuhai), Zhuhai,
519082, China
[5] State Key Laboratory of Organic Geochemistry and Guangdong Key Laboratory of
Environmental Protection and Resources Utilization, Guangzhou Institute of
Geochemistry, Chinese Academy of Sciences, Guangzhou 510640, China
[6] CAS Center for Excellence in Deep Earth Science, Guangzhou, 510640, China
[7] University of Chinese Academy of Sciences, Beijing 100049, China
[8] Guangzhou Ecological and Environmental Monitoring Center of Guangdong Province,
Guangzhou 510060, China
[9] Institute of Mass Spectrometer and Atmospheric Environment, Jinan University,
Guangzhou 510632, Guangdong, China
[10] Guangdong Provincial Key Laboratory of Optical Fiber Sensing and
Communications, and Department of Optoelectronic Engineering, Jinan University,
Guangzhou, 510632, China
* Corresponding authors: byuan@jnu.edu.cn



## Abstract

Volatile organic compounds (VOCs) are key precursors of ozone and particulate matter that are the two dominant air pollutants in urban environments. However, compositions and sources of VOCs in urban air aloft were rarely reported by far. To address this matter, highly time-resolved measurements of VOCs were made by proton-transfer-reaction time-of-flight mass spectrometer (PTR-ToF-MS) at a 450-m platform on the Canton Tower in Guangzhou, China. A combination of *in-situ* measurements and modeling techniques was used to characterize variations and sources of VOCs. Five sources were identified from positive matrix factorization (PMF) analysis, namely daytime-mixed (e.g., biogenic emissions and secondary formation), visitor-related (e.g., human breath and volatilization of ethanol-containing products), vehicular+industrial, regional transport, and volatile chemical product (VCP)-dominated (i.e., volatilization of personal care products), contributing on average to 22%, 30%, 28%, 10%, and 11% of total VOC (TVOC) mixing ratios, respectively. We observe that contributions of the visitor-related source, mainly composed of ethanol, followed well with the variation patterns of visitor number on the tower. The VCP-dominated source only had an average contribution of ~5.7 ppb during the campaign, accounting for a small fraction (11%) of TVOC mixing ratios. However, large fractions of some VOC species, e.g., monoterpenes (49%), were attributed to the VCP-dominated source, indicating significant contributions of VCPs to ambient concentrations of these species in urban environments. Vertical profiles of air pollutants (including NOx, ozone, Ox, and $PM_{2.5}$), measured at 5 m, 118 m, 168 m, and 488 m, exhibited more evident gradients at night than in the daytime owing to stronger stability of the nocturnal boundary layer. Mixing ratios of VOC species during the nighttime generally decreased with time when the 450-m platform was located in the nocturnal residual layer and significantly increased when impacted by emissions at ground. The results in this study demonstrated composition characteristics and sources of VOCs in urban air aloft, which could provide



valuable implications in making control strategies of VOCs and secondary air pollutants.


# 1 Introduction

Volatile organic compounds (VOCs) are important trace gases in the atmosphere and are composed of myriad chemical species (Pallavi et al., 2019; Wang et al., 2020a; Gkatzelis et al., 2021). Except for their direct adverse impacts on human health (Zhang et al., 2013), VOCs also contributed significantly to the formation of secondary pollutants such as ozone and secondary aerosol (Vo et al., 2018; Zhou et al., 2019; Qin et al., 2021). Reduction in ambient VOCs concentrations is the key for synergistic control of both ozone and particle pollution. However, it is highly challenging for this target due to complex sources and chemical transformations of VOCs in urban environments (Yuan et al., 2012; Mo et al., 2016; Zhu et al., 2019).

In addition to compiling accurate emission inventories (bottom-up method) (Zheng et al., 2013; An et al., 2021), the combination of *in-situ* measurements and receptor models (top-down method) was widely adopted to quantitatively apportion sources of ambient VOCs (Baudic et al., 2016; Liu et al., 2016; Fan et al., 2021; Pernov et al., 2021). Concentrations of various VOC species could be measured by offline and online techniques. Gas chromatography-flame ionization detector/mass spectrometry (GC-FID/MS) combined with stainless steel canisters were the most popular offline technique for VOCs measurements (Guo et al., 2011; Yuan et al., 2013; Zhang et al., 2013; Qin et al., 2021). Automated online GC-FID system and high time resolution mass spectrometer, such as proton-transfer-reaction mass spectrometer (PTR-MS) and chemical ionization mass spectrometer (CIMS), were popular online techniques for VOCs measurements (de Gouw and Warneke, 2007; Wang et al., 2020a; Wang et al., 2020c; Fan et al., 2021; Ye et al., 2021). However, VOCs measurements made by both online and offline instruments are significantly affected by very local emission sources, particularly in urban environments, when they are usually deployed at ground level. This is highly important for studies aiming to characterize variations and sources of ambient VOCs at large spatial scales (such as a city or city clusters) based on



measurements of only one site. To address this concern, VOCs measurements made in
the upper part of the planetary boundary layer (PBL) may be a better choice due to the
well mixing of surface emissions when being transported upward from sources to
observation sites (Hu et al., 2015a; Hu et al., 2015b; Squires et al., 2020).

As reported in the literature, *in-situ* measurements of VOCs at high altitudes (e.g.,

hundreds of meters or several kilometers above ground level) were predominantly made
using the combination of offline techniques and samples collected by various platforms
such as aircraft (Geng et al., 2009; Xue et al., 2011; Benish et al., 2020), tethered
balloons (Zhang et al., 2018; Wu et al., 2020b; Wang et al., 2021; Wu et al., 2021), high
buildings and towers (Ting et al., 2008; Mo et al., 2020), and unmanned aerial vehicles
(UAVs) (Vo et al., 2018; Liu et al., 2021). These offline measurements were
predominantly used to reveal vertical variations of VOCs concentrations, impacts of
VOCs degradation chemistry on the formation of secondary pollutants, and source
characteristics of the species of interest. Offline measurements made at high altitudes
were generally not capable of fully characterizing temporal variations of concentrations
and source characteristics of VOCs due to strict limitations in their time resolution and
sample sizes. In this condition, online VOCs measurements with fast response at high
altitudes are required. Lack of available platforms has been a key limited factor for
conducting online VOCs measurements at high altitudes in China. For instance, the
combined utilization of aircraft and online spectrometer (such as PTR-MS) has been
widely used in North America to measure VOCs concentrations in the lower
troposphere (Hornbrook et al., 2011; Müller et al., 2016; Yuan et al., 2016; Koss et al.,
2017; Fry et al., 2018; Chen et al., 2019), while it is quite difficult in China due to the
lack of professional research aircraft and the strict control of airspace. Tethered balloons
and UAVs are generally not suitable for online VOCs measurements due to their limited
payloads (Dieu Hien et al., 2019). Tower-based platforms provide another path for
online VOCs measurements at high altitudes in urban environments. However, tower-
based online measurements of VOCs were only reported in Beijing, China by far



(Squires et al., 2020; Zhang et al., 2020).
In this study, continuous online VOCs measurements, including more than 200
chemical species with a time resolution of 10 s, were made at a 450-m platform on the
Canton Tower in the Pearl River Delta (PRD) region, China during August–November
2020. A combination of the measurements and the positive matrix factorization (PMF)
receptor model was used to provide new insights into the concentrations, temporal
variations, and source contributions of VOCs in urban region.

## 2    Methods and materials

### 2.1 Site description and field campaign

The PRD region is one of the most developed city clusters in China with more
than 70 million residents by 2020 and is suffering from air pollution problems (e.g.,
ozone and secondary aerosol) (Wang et al., 2017; Wang et al., 2020b; Yan et al., 2020;
Li et al., 2022). In this study, VOCs measurements were made at the Canton Tower
(CTT, 23.11°N, 113.33°E) in Guangzhou, a large city in PRD (Figure S1), from August
18 to November 5 in 2020. The CTT has a total height of 610 m including the shaft on
the top. The observation was conducted in a room (Figure S1) at the 450-m Look Out
platform (Jin et al., 2022). The 450-m Look Out platform is a famous tourist attraction
with an opening time of local time (LT, UTC+8) 10:00–22:30, and visitors could walk
around for a panorama of downtown Guangzhou. On each day, there are two busiest
tourist hours, roughly at LT 11:00-14:00 and 18:00-21:00, on the 450-m platform. In
addition, there are three restaurants between 376 and 423 m. The VOCs measurements
were interrupted during October 8–12 due to instrument malfunction.

### 2.2 VOCs measurements

VOCs measurements were made using a high-resolution proton-transfer-reaction
quadrupole interface time-of-flight mass spectrometer (PTR-QiToF-MS, Ionicon
Analytik, Innsbruck, Austria) with both hydronium ion ($H_3O^+$) and nitric oxide ion





(NO$^+$) chemistry (Yuan et al., 2017; Wang et al., 2020a; Wu et al., 2020a). The H$_3$O$^+$
and NO$^+$ modes were automatically switched with 22 min for the H$_3$O$^+$ mode and 12
min for the NO$^+$ mode during the campaign. In this study, only VOCs measurements
made in the H$_3$O$^+$ mode were used for analysis. The sampling inlet of the instrument
was extended to the outside wall of the observation room using a ~5-m long
Perfluoroalkoxy (PFA) Teflon tubing (OD: 1/4"), which is drawn by a pump at a flow
rate of ~5 L min$^{-1}$. Blank measurements were performed automatically at the last 2 min
of the H$_3$O$^+$ mode by passing ambient air through a platinum catalyst heated to 365 ℃.
Mass spectra of up to $m/z$ = 510 were obtained at a time resolution of 10 s.

A gas standard with 35 VOC species (Table S1) was used for calibrations of the

PTR-ToF-MS once per day. Ten organic acids and nitrogen-containing VOC species
were calibrated using a liquid calibration unit in the laboratory (Table S1). Sensitivities
of the remaining VOC species were determined using the quantification method based
on reaction kinetics of the PTR-ToF-MS (Wu et al., 2020a; He et al., 2022). Impacts of
the change in ambient humidity on measured signals of the PTR-ToF-MS were removed
using humidity-dependence curves of VOC species determined in the laboratory (Wang
et al., 2020a; Wu et al., 2020a). The limit of detection (LOD) for a VOC species was
defined as the concentration when the signal-to-noise ratio (SNR) equals to 3 (Yuan et
al., 2017). Average mixing ratios, LOD, sensitivities, chemical formula, and suggested
compounds of 225 VOC species used in this study are summarized in Table S1.

### 156  2.3 Other measurements

During the CTT campaign, a CO$_2$ and H$_2$O Gas Analyzer (Model: Li-840A, Licor

Inc., USA) was deployed to measure carbon dioxide (CO$_2$, ppm) and humidity (mmol
mol$^{-1}$). In addition, four air quality automatic monitoring stations are located at ground
level (~5 m), 118 m, 168 m, and 488 m of the tower, which report hourly concentrations
of O$_3$, NO, NO$_2$, NOx, and PM$_{2.5}$ along with meteorological parameters including
temperature (T), relatively humidity (RH), and pressure (Mo et al., 2020). Mass
concentrations of gaseous pollutants were reported at 25 ℃ and 1013.25 hPa and were



converted to mixing ratios (ppb) accordingly. A ceilometer (CL31, Vaisala, Finland)
deployed on the Panyu Campus of Jinan University (23.02°N, 113.41°E, Figure S1),
approximately 13.5 km to the southeast of the CTT, was used to measure planetary
boundary layer height (PBLH) during the camapign. In addition, measurements of
VOCs and $CO_2$ made on the campus of Guangzhou Institute of Geochemistry (GIG),
Chinese Academy of Sciences (23.15°N, 113.36°E, ~25 m above ground level) during
September–November 2018 (Wang et al., 2020a; Wang et al., 2020c; Wu et al., 2020a)
were used for comparison with those measured on the CTT. The GIG site is located
approximately 5.7 km to the northeast of the CTT. Measurements of VOCs and $CO_2$ at
the GIG site were made using the same instruments as those at the CTT site.
**2.4 PMF receptor model**
The PMF receptor model was used to quantitatively analyze sources of the VOC
measurements made at the 450-m platform. The PMF model has been widely used to
determine source contributions of measured VOCs concentrations in previous studies
(Yuan et al., 2012; Pallavi et al., 2019; Pernov et al., 2021). A simple description of the
PMF model was provided in the Supplementary Information (SI).
The PMF model was performed on 225 VOC species in this study (Table S1). In
preparation of PMF input data, measured concentrations of a VOC species below the
LOD were replaced with half of the LOD and corresponding uncertainties were
assigned to 5/6 of the LOD. Missing samples of a VOC species were replaced with its
median value during the campaign and corresponding uncertainties were set as values
equal to three times the median value (Zhang et al., 2013; Pernov et al., 2021; Qin et
al., 2021). In this study, the measured ethanol concentrations on the 450-m platform
were significantly impacted by the change in the number of visitors (a detailed
discussion in Section 3.3) and exhibited strong variations during the campaign (Figure
1). Thus, measurement uncertainties of ethanol calculated by Eq. (S3) were reduced by
a factor of 5 to increase the weight of ethanol in PMF analysis, which successfully
resolved factors representing visitor influences and significantly reduce residuals of





PMF solution from over 20% to ~14%. The PMF analysis was performed using the
PMF Evaluation Tool (v3.05) with Igor Pro (Ulbrich et al., 2009).

## 3    Results and discussion

### 3.1    Overview of field measurements during the campaign

As shown in Figure 1, concentrations of various species and meteorological
parameters all exhibited strong variations during the campaign. Daily mean ozone
mixing ratios varied in the range of 17.8–105.0 ppb with an average (± standard
deviation) of 55.1 ± 18.3 ppb. Daily mean total VOC (TVOC) mixing ratios, including
a total of 225 species, varied between 23.9–124.2 ppb with an average of 62.1 ± 21.8
ppb. Daily mean NOx mixing ratios varied in the range of 7.9–31.6 ppb with an average
of 13.6 ± 3.8 ppb. Measured $CO_2$ mixing ratios exhibited strong variability with daily
mean values ranging from 403.5 to 471.4 ppm. Ethanol was the most abundant VOC
species, accounting on average for 23.5% of measured TVOC mixing ratios during the
campaign. Daily mean ethanol mixing ratios varied between 4.3–53.4 ppb with an
average of 15.3 ± 9.1 ppb. Toluene was the most abundant aromatic species and had an
average mixing ratio of 1.4 ± 0.9 ppb during the campaign. Daily mean temperatures
varied in the range of 17.7–29.0 ℃ with an average of 23.2 ± 3.0 ℃. Daily mean RH
varied between 39.3%–85.0% with an average of 71.6% ± 10.3%. In general, the
observation site was predominantly influenced by hot and moist air masses from August
18 to October 4, but cooler and dryer air masses from October 5 to November 5.
The most abundant 10 VOC species measured by PTR-ToF-MS during the
campaign were ethanol, methanol, acetic acid, formaldehyde, acetone, ethyl acetate,
acetaldehyde, hydroxyacetone+propionic acid, toluene, and C8 aromatics, contributing
to over 70% of TVOC mixing ratios. As shown in Figure 2, the 225 VOC species were
classified into six categories, namely $C_xH_y$ (i.e., hydrocarbons), $C_xH_yO_1$ (i.e., VOC
species containing one oxygen atom), $C_xH_yO_2$ (i.e., VOC species containing two
oxygen atoms), $C_xH_yO_{\geq 3}$ (i.e., VOC species containing more than three oxygen atoms),





N/S containing species (i.e., VOC species containing nitrogen or sulfur atoms), and
siloxanes (Wu et al., 2020a; He et al., 2022). The most abundant category was $C_xH_yO_1$,
which had an average contribution of 62.2% to measured TVOC mixing ratios. The
$C_xH_yO_2$ and $C_xH_yO_{\geq 3}$ categories contributed to 24.9% and 2.9% of measured TVOC
mixing ratios, respectively. $C_xH_y$ was the third abundant category, accounting for 6.4%
of measured TVOC mixing ratios. Concentrations of N/S containing species and
siloxanes were generally lower than 0.5 ppb and only contributed to 1.3% and 2.4% of
measured TVOC mixing ratios, respectively.

As shown in Figure 2, the majority of the $C_xH_y$, $C_xH_yO_3$, and N/S containing

species measured at 450 m (CTT campaign) had lower mixing ratios than those
measured at ground level (GIG campaign), indicating their predominant contributions
from surface emission sources. Most of the $C_xH_yO_1$ and $C_xH_yO_2$ species measured at
450 m had comparable mixing ratios to those measured at the ground level. However,
mixing ratios of some $C_xH_yO_2$, $C_xH_yO_3$, and N/S containing species measured at 450 m
were significantly higher than at the ground level, which can be attributable to either
enhancement of their emissions on the 450-m platform or more secondary formation
from oxidation of VOCs (e.g., $C_xH_y$ and $C_xH_yO_1$ species).

## 3.2   Diurnal variations in selected VOC species

Average diurnal profiles of nine selected VOC species measured by PTR-ToF-MS

during the campaign are demonstrated in Figure 3. Measurement results at GIG in 2018
are also shown for comparison to investigate differences in their diurnal variation
patterns and likely sources when measured at ground level and in urban upper air. In
addition, average diurnal profiles of the selected VOC species on working and non-
working days (including weekends and public holidays when the 450-m platform had
more visitors) during the CTT campaign are compared to explore potential emissions
from visitors. Meteorological factors including temperature and RH exhibited
insignificant differences between working and non-working days (Figure S2). Thus,
differences of concentrations between working and non-working days for various





species were not caused by the change in meteorological conditions.
Diurnal profiles of aromatic species, including benzene, toluene, and C8 aromatics
measured at 450 m exhibited similar variability with minima occurring between LT
12:00–16:00. Aromatics with higher chemical reactivity could be removed more rapidly
by reactions with hydroxyl radicals (OH) in the daytime (Yuan et al., 2012; Wu et al.,
2020a). In addition, significant elevation of daytime PBL could enhance the dilution of
chemical species, leading to rapid decreases in their concentrations (Sangiorgi et al.,
2011; Zhang et al., 2018). The two effects are the two most important factors for
controlling diurnal profiles of aromatics measured at 450 m. By contrast, diurnal
profiles of aromatics measured at ground displayed a different pattern with two peaks
occurring in the morning (LT 07:00–08:00) and evening (LT 19:00–22:00), respectively.
Diurnal patterns of aromatics are highly consistent with that of NOx (a typical tracer of
traffic emissions in urban region) at the ground level but were significantly different
from that of NOx at 450 m (Figure 4). Therefore, measured concentrations of aromatics,
particularly for benzene, were significantly affected by traffic emissions at ground level,
but contributed by more complex sources at 450 m. The diurnal profiles of aromatics
on working and non-working days exhibited minor differences, implying insignificant
contributions from visitor-related emissions. On working days, toluene concentrations
measured at 450 m were more affected by traffic emissions as manifested by the two
significant peaks in the morning and late afternoon.
Isoprene and monoterpenes exhibited distinct diurnal variation patterns during the
two campaigns. As reported in (Gómez et al., 2020; Tan et al., 2021), diurnal profiles
of isoprene and monoterpenes concentrations in non-urban regions usually displayed
unimodal patterns with a peak occurring at noon due to the strong light/temperature-
dependence of biogenic emissions. In this study, isoprene concentrations at 450 m
plateaued during the daytime and were slightly higher on non-working days than those
on working days, implying significant contributions from visitor-related emissions. The
diurnal profile of monoterpenes measured at 450 m exhibited a bi-modal pattern with





two peaks at LT 14:00 and 20:00, which was roughly in accordance with diurnal peaks
of visitor numbers on the 450-m platform. In addition, monoterpenes concentrations at
450 m were significantly higher on non-working days (particularly during the busiest
tourist hours) than on working days, confirming significant contributions from visitor-
related or cooking emissions (Klein et al., 2016). The diurnal profiles of methyl vinyl
ketone (MVK) + methacrolein (MACR) demonstrated similar shapes to ozone at both
450 m and ground level with maxima occurring between LT 13:00–15:00 (Figure 4),
consistent with MVK+MACR as photooxidation products of isoprene (Greenberg et al.,
1999; Zhao et al., 2021). The concentrations of MVK+MACR during the daytime on
non-working days were also evidently higher than those on working days, which are
consistent with isoprene observations.
Acetone, methanol, and ethanol are abundant OVOC species in urban atmosphere.
Diurnal profiles of acetone measured at both 450 m and the ground level were
characterized by higher concentrations in the daytime, suggesting significant
contributions from daytime sources, such as vegetation emissions and photooxidation
of hydrocarbons (Hu et al., 2013; Gkatzelis et al., 2021). In addition, acetone
concentrations at 450 m were higher on non-working days than on working days,
implying prominent contributions from visitor-related emissions. Diurnal profiles of
methanol and ethanol measured at ground level were characterized by a bimodal pattern
with two peaks occurring in the morning (LT 08:00) and evening (LT 20:00),
respectively, confirming significant contributions from traffic emissions. However,
methanol concentrations measured at 450 m exhibited insignificant diurnal variability
and lower concentrations on non-working days, indicating that they were less affected
by visitor-related emissions. The diurnal profile of ethanol at 450 m displayed two
peaks at LT 13:00 and 19:00, respectively, which was in accordance with the two busiest
tourist hours of the 450-m platform. In addition, ethanol concentrations at 450 m on
non-working days were significantly higher than those on working days, particularly in
the opening hours of the 450-m platform. These results suggest that the ethanol



concentrations measured at 450 m were largely contributed by visitor-related emissions.
To further explore spatial scales of emission source regions for different VOC
species, autocorrelation profiles of their time series were calculated by offsetting time
from -120 to 120 min. As indicated in previous studies (Hayes et al., 2013; Hu et al.,
2016), species more affected by local sources would have a narrower autocorrelation
profile. As shown in Figure 4, peak widths of autocorrelation profiles for different
chemical species at 450 m varied significantly. Autocorrelation profiles of
monoterpenes, toluene, ethanol, methanol, and isoprene were relatively narrower (even
narrower than the autocorrelation profile of NOx), and thus sources of these species
had more local characteristics. Autocorrelation profiles of benzene, C8 aromatics,
acetone, and MVK+MACR were much flatter (but narrower than the autocorrelation
profile of ozone and Ox), indicating that concentrations of these species were more
contributed by sources at larger spatial scales. By contrast, peak widths of the
autocorrelation profiles for different chemical species (except for ethanol) varied
insignificantly at ground level and were similar to that of NOx. Therefore,
concentrations of the selected VOC species were significantly contributed by local
traffic emissions at ground level but contributed by more complex sources on larger
spatial scales at 450 m.
**3.3  Impacts of visitor-related emissions on VOCs measurements**
As introduced in section 2.1, the CTT campaign was conducted in August-
November of 2020, during which visitors were required to wear masks when visiting
the CTT and ethanol-containing products were widely used to prevent the spread of the
COVID-19 pandemic. For example, medicinal alcohol (75%) spray was widely used to
wipe public utilities and 75%-ethanol bacteriostatic gel was extensively used as
sanitizer for hands. Diurnal profiles of some VOC species (e.g., ethanol and
monoterpenes) exhibited similar diurnal patterns to the number of visitors at the 450-m
platform. Therefore, VOCs measurements made at the 450-m platform may be affected
by visitor-related emissions, such as human breath and evaporation of personal care


products (Veres et al., 2013).

As shown in Figure 5, the diurnal profile of $CO_2$ measured at 450 m increased

between LT 09:00–20:00, which was different from those measured at ground level.
The higher $CO_2$ mixing ratios at 450 m were predominantly contributed by human
breath due to the absence of combustion sources. Measured ethanol mixing ratios were
well correlated with those of $CO_2$ ($r$=0.62) during the CTT campaign, indicating that
ethanol concentrations, as well as its variations, were predominantly determined by the
change in the number of visitors on the tower. In addition, the $CO_2$ mixing ratios on
non-working days, especially during the busiest tourist hours, were markedly higher
than those on working days. As illustrated in Figure 5, the 450-m platform was closed
during October 13-15 as the result of the influence of Typhoon Kompasu. On these days,
mixing ratios of ethanol, $CO_2$, and monoterpenes exhibited similar variation patterns to
benzene (a typical tracer of traffic emissions). However, mixing ratios of ethanol, $CO_2$,
and monoterpenes exhibited quite different variation patterns from benzene when the
450-m platform was re-open (October 16–21). For instance, mixing ratios of ethanol,
$CO_2$, and monoterpenes generally decreased from LT 12:00 to 18:00 between October
13–15, but significantly increased during the same period between October 16–21.
Therefore, it can be concluded that the VOCs measurements made at the 450-m
platform were significantly affected by visitor-related emissions, which will be
quantitatively assessed using the PMF analysis in following sections.

### 3.4   Source analysis of VOCs measurements

In this study, a five-factor solution for the PMF analysis was chosen as the optimal

result. Figure 6 displays source profiles (m/z ≤ 150, the full range of the mass spectra
is shown in Figure S5) of the five PMF factors along with average diurnal profiles of
their contributions. The five factors were assigned to likely sources of daytime-mixed,
visitor-related, vehicular+industrial, regional transport, and volatile chemical product
(VCP)-dominated according to characteristics of their source profiles and temporal
variations, which are detailed discussed in the *SI*.





The visitor-related source predominantly includes contributions from human
breath and volatilization of personal care products. Contributions of the visitor-related
source had the narrowest autocorrelation profile among the five factors (Figure 6),
confirming its most local characteristics. As shown in Figure 7, the visitor-related
source had the largest contributions (15.9 ± 19.6 ppb), accounting for 30% of the
average TVOC mixing ratio. In addition, contributions of the visitor-related source
accounted for a larger fraction of TVOC mixing ratios on non-working days (33%) than
those on working days (28%). The vehicular+industrial source mainly includes
contributions from vehicular exhausts and emissions of various industrial processes.
Contributions of the vehicular+industrial source (15.1 ± 18.3 ppb) were comparable to
those of the visitor-related source, accounting for 28% of the average TVOC mixing
ratio. As also anticipated, the vehicular+industrial source contributed to a smaller
fraction of TVOC mixing ratios on non-working days (26%) than those on working
days (30%). The VCP-dominated source predominantly includes contributions from
VCPs in urban environments. The VCP-dominated source had an average contribution
of 5.7 ± 5.4 ppb, accounting for 11% of the average TVOC mixing ratio. The average
contribution of the VCP-dominated source in this study was comparable to those (~6.0
ppb) measured in New York City (Gkatzelis et al., 2021). However, VCPs contributed
to over 50% of anthropogenic VOCs emissions in New York City, which is significantly
greater than the fraction in this study (11%, and it will increase to 16% when
contributions of the visitor-related source were removed). In comparison to large cities
in U.S., traffic and industrial emissions were still dominant sources of ambient VOCs
in Chinese cities.
The daytime-mixed source predominantly includes contributions from biogenic
emissions and photooxidation products of various VOCs. As shown in Figure 7, the
daytime-mixed source had an average contribution of 11.6 ± 12.6 ppb, accounting for
21% of the average TVOC mixing ratio. It exhibited consistent diurnal variation
patterns on both working and non-working days but had larger contributions in the


daytime on non-working days (Figure 6). This may be attributed to the enhanced
formation of secondary OVOC species as manifested by the higher ozone
concentrations on non-working days (Figure S6). The regional transport source mainly
includes contributions from advection transport of aged air masses. Contributions of
the regional transport source had the flattest autocorrelation profile (Figure 6), implying
its most regional characteristics. Only a small fraction (<5%) of reactive chemical
species such as aromatics were attributed to this factor. Contributions of the regional
transport source accounted for 13% of the TVOC mixing ratio when affected by
continental airflows, but only accounted for 3% when affected by marine airflows
(Figure S7). By contrast, contributions of the other factors displayed insignificant
dependence on wind direction.
As shown in Figure 8, source apportionment of the selected VOC species (Figure
3) discussed in section 3.2 were further investigated. The vehicular+industrial source
had the largest contribution (36%) to benzene. The daytime-mixed source also
contributed to 18% of measured benzene mixing ratios. In addition, more than 20% of
benzene was attributed to the VCP-dominated source. In contrast to benzene, toluene
was predominantly attributed to the vehicular+industrial (93%) and visitor-related (7%)
sources. The average ratio of toluene to benzene was 5.7 ppb/ppb during the CTT
campaign (Figure S8), further confirming primary contributions of toluene from
vehicular and industrial emissions (Wu et al., 2016; Zhou et al., 2019; Xia et al., 2021).
The vehicular+industrial source also accounted for the largest fractions of C8 and C9
aromatics. In addition, 26% of C8 aromatics and 38% of C9 aromatics were attributed
to the VCP-dominated source. The other three sources in total contributed to less than
10% of concentrations of C8 and C9 aromatics. These results indicate that VCPs are
important sources of aromatics in urban environments but they were rarely identified
in previous studies.
Isoprene and monoterpenes are widely known tracers of biogenic emissions
(Millet et al., 2016; Zhao et al., 2021). However, the daytime-mixed source only



contributed to 16% of measured isoprene mixing ratios. By contrast, more than 70% of
isoprene were attributed to the visitor-related (38%) and VCP-dominated (35%) sources.
As for monoterpenes, more than 95% of the measured mixing ratios were attributed to
the visitor-related (47%) and VCP-dominated (49%) sources. The average ratio of
monoterpene to isoprene mixing ratios at 450 m was 0.84 in the daytime (LT 08:00–
18:00), which was significantly greater than that at the ground level (0.05) (Figure S8).
It further confirms significant contributions of monoterpenes from visitor-related
emissions at the 450-m platform. The daytime-mixed source did not exhibit discernible
contributions to monoterpenes. This agrees well with the results in New York City
where monoterpene mixing ratios were primarily attributed to anthropogenic sources
such as VCPs, cooking, and building materials (Coggon et al., 2021; Gkatzelis et al.,
2021). These results suggest that emission intensities of isoprene and monoterpenes
may be highly underestimated in urban regions if their anthropogenic emissions are
overlooked or less considered. This is exceedingly important for air quality models
when estimating formation of ozone and secondary organic aerosol driven by the
oxidation of isoprene and monoterpene. As the key photooxidation products of isoprene,
nearly 60% of MVK+MACR were attributed to the daytime-mixed source. The visitor-
related, regional transport, and VCP-dominated sources contributed to comparable
fractions (11%–15%) of MVK+MACR. Therefore, anthropogenic emissions are also
important sources of MVK+MACR in urban environments.

As shown in Figure 8, 39% of acetone was attributed to the daytime-mixed source.

The vehicular+industrial (19%) and VCP-dominated (21%) sources accounted for
comparable fractions of measured acetone mixing ratios. In addition, the visitor-related
source also contributed (7%) significantly to acetone. As for methanol, the
vehicular+industrial source accounted for the largest fraction (38%), followed by the
daytime-mixed (22%), regional transport (17%), VCP-dominated (14%), and visitor-
related (9%) sources. These results reveal that VCPs also contributed significantly to
ambient concentrations of acetone and methanol and should be carefully considered



when estimating their total emission intensities from anthropogenic sources. Ethanol
was predominantly attributed to the visitor-related source. Therefore, the enhanced
ethanol mixing ratios were not capable of representing its characteristic concentrations
in urban environments. Although the absence of synchronous ground-level
measurements, we can speculate that ethanol concentrations at ground level were also
significantly increased during the outbreak of the COVID-19 pandemic due to the
extensive usage of ethanol-containing products. The enhancement of ethanol
concentrations may contribute significantly to atmospheric OH reactivity (Millet et al.,
2012; de Gouw et al., 2017; de Gouw et al., 2018) and then regulate the formation of
secondary pollutants. Therefore, impacts of the ethanol enhancement on ambient air
quality should be explicitly investigated in future studies due to the wide report of ozone
enhancement during the outbreak of the COVID-19 pandemic (Huang et al., 2020; Qi
et al., 2021).
Acetonitrile is widely used as a typical tracer of biomass burning sources in
previous studies (de Gouw et al., 2003; Zhang et al., 2020; Tan et al., 2021). However,
biomass burning source was not identified in this study because acetonitrile was not
predominantly attributed to a single factor (Figure 8). In addition to the visitor-related
source, the other four sources also had significant contributions to acetonitrile. As
indicated by (Huangfu et al., 2021), it is not always suitable, particularly in urban
environments, to use absolute concentrations of acetonitrile as the indication of biomass
burning sources. The ratio of acetonitrile to CO is a better indicator to identify whether
VOC measurements are significantly contributed by biomass burning emissions. The
average ratio of acetonitrile to CO was only 0.09 (ppb ppm$^{-1}$) during the campaign
(Figure S8), indicating insignificant contributions from biomass burning sources. In
addition to the daytime-mixed (22%) and vehicular+industrial (26%) sources, the VCP-
dominated source (31%) was also an important source of acetonitrile in urban
environments.



### 3.5 Vertical distributions of air pollutants concentrations


As introduced in section 2.1, hourly concentrations of some air pollutants were
routinely measured at four automatic sites on the CTT. Figure 9 shows time series of
vertical profiles of NOx, ozone, Ox ($O_3+NO_2$), and $PM_{2.5}$ concentrations in September
2020. Concentrations of the four pollutants all exhibited significantly stratified
structures between 488 m and the ground level. Higher mixing ratios of ozone and Ox
predominantly occurred at higher altitudes, while higher NOx mixing ratios mainly
occurred at ground level. By contrast, higher $PM_{2.5}$ concentrations were observed at
both middle altitudes and ground level.
To further clarify vertical distribution patterns of air pollutants concentrations,
their composite profiles for daytime (LT 08:00–18:00), nighttime (LT 19:00–05:00),
and the whole day in the campaign were determined, respectively, as shown in Figure
10. Vertical profiles of air pollutants concentrations exhibited similar shapes both in
daytime and nighttime. NOx mixing ratios decreased from the ground level to 488 m,
suggesting intensive surface emissions around the CTT. Ozone mixing ratios rapidly
increased from the ground level to 488 m, which was consistent with the results reported
in previous studies (Velasco et al., 2008; Li et al., 2018; Zhang et al., 2019; Li et al.,
2021b). The positive gradients of ozone profiles are mainly caused by enhanced NO
titration ($NO+O_3=O_2+NO_2$) and dry deposition near ground. Ox mixing ratios also
increased from the ground level to 488 m but exhibited weaker gradients in comparison
to ozone. Vertical profiles of $PM_{2.5}$ concentrations exhibited similar shapes to NOx
during the campaign. Daily mean concentrations of $PM_{2.5}$ and Ox were well correlated
at the four altitudes with $r$ values varying in the range of 0.61–0.82, suggesting
prominent contributions of secondary formation to ambient PM concentrations.
Moreover, the correlation coefficients between Ox and $PM_{2.5}$ concentrations at 488 m
(0.82) were greater than those at ground level (0.78), as they were less affected by
nearby vehicular emissions. This is consistent with the work by (Yan et al., 2020), who
reported that secondary components contributed to ~80% of $PM_{2.5}$ concentrations in



PRD over the 2008–2019 period.

As shown in Figures 9 and 10, vertical profiles of air pollutants concentrations

exhibited weaker gradients in the daytime than in the nighttime. Therefore, the daytime
VOC chemistry may have minor differences between the ground level and the 450-m
site due to strong vertical mixing of chemical species in the planetary boundary layer
(PBLH>450 m, as shown in Figure S9). In the nighttime, the oxidative products (such
as organic nitrates and OVOCs) of unsaturated hydrocarbons, predominantly initiated
by nitrate radicals ($NO_3$) and ozone, are also important precursors of secondary aerosol
(Warneke et al., 2004; Brown et al., 2011; Ng et al., 2017; Liebmann et al., 2019).
However, it is highly challenging to investigate the nighttime VOC chemistry with only
ground-level measurements due to the rapid removal of $NO_3$ radicals and ozone by
enhanced NO titration in the near-surface atmosphere (Geyer and Stutz, 2004; Stutz et
al., 2004; Brown et al., 2007). In this condition, the nocturnal residual layer, separated
from nocturnal boundary layer and remained, to a large extent, the chemical
composition of the daytime atmosphere, could provide an ideal place for investigating
nighttime VOC chemistry. Oxidative products of VOCs in the residual layer could be
mixed downward with the expansion of the PBL during the daytime (Geyer and Stutz,
2004; Stutz et al., 2004; Li et al., 2021a), contributing to the formation of ozone and
secondary aerosol at ground level. Investigation of the nighttime VOC chemistry was
one of the initial purposes of this study. Unfortunately, the 450-m site was rarely located
in the nocturnal residual layer during the campaign due to frequent occurrences of
cloudy and rainy weather. The average nighttime PBLH in Guangzhou was
approximately stabilized at 500 m during the campaign (Figure S9), implying
significant impacts from surface emissions on the measurements made at 450 m.

In addition to the measured PBLH data, formation of the residual layer at 450 m

could be also identified by comparing differences of ozone mixing ratios between 488
m and the ground level. Without fresh NO emissions, ozone mixing ratios in the
nocturnal residual layer were markedly higher than at ground level and exhibited





insignificant variability throughout the nighttime (Caputi et al., 2019; Udina et al.,
2020). By contrast, surface ozone mixing ratios are generally very low (close to zero)
due to enhanced titration by freshly emitted NO and strong inhibition of atmospheric
vertical mixing (Ma et al., 2011; Chen et al., 2020). In this study, the data collected
between September 27–30 was one of the cases discussed above and was used to briefly
describe behaviors of some representative VOC species (including ethanol,
monoterpene, styrene, phenol, and toluene) at 450 m.
As shown in Figure 11, ozone mixing ratios measured at ground level ($10.2 \pm 10.4$
ppb) were significantly lower than those at 488 m ($44.2 \pm 19.6$ ppb) on the nighttime
of September 27–30, indicating formation of the nocturnal residual layer lower than
450 m. On the nighttime of September 27–28, ozone mixing ratios at 488 m slightly
fluctuated around 46.8 ppb between LT 19:00–00:00 and suddenly decreased to 28.4
ppb at LT 01:00 on September 28. The sudden decrease in ozone at 488 m at LT 01:00
was accompanied by slight increases in both NOx and VOCs but significant decreases
in NOx and NO at ground level, indicating a transitory intrusion of surface fresh
emissions into the residual layer. On September 28, ozone mixing ratios at 488 m
slightly decreased from 33.0 to 31.5 ppb from LT 02:00 to 05:00, during which mixing
ratios of NOx and VOCs all decreased in different degrees. The continuous decreases
in both toluene and ethanol between LT 02:00–05:00 confirm that the VOCs
measurements at 450 m were free of interferences by fresh emissions due to their
significant contributions from vehicular exhausts (Figure 7). Toluene mixing ratios
decreased by 43% from LT 02:00 to 05:00, which was significantly larger than those
(12–27%) of the other VOC species shown in Figure 11. However, the $NO_3$ reactivity
(characterized by reaction rate constants of VOC species to $NO_3$ radical, $k_{NO3}$) of
toluene ($k_{NO3} = 7 \times 10^{-17}$ cm$^{-3}$ molecule$^{-1}$ s$^{-1}$) is exceedingly lower than those of the other
unsaturated VOC species ($k_{NO3}$ varies in the magnitudes of $10^{-12}$ cm$^{-3}$ molecule$^{-1}$ s$^{-1}$)
(Atkinson and Arey, 2003; Atkinson et al., 2006). Therefore, the decline of unsaturated
VOC species in the nocturnal residual layer may not be all attributed to the degradation





chemistry initiated by $NO_3$ radicals or ozone.

On the nighttime of September 28–29, the PBLH was higher than 500 m between

LT 19:00–00:00, resulting in significant decreases in ozone and increases in NOx and
VOCs. As shown in Figure 11, the 450-m site may locate in the residual layer after LT
01:00. However, the rapid decrease in mixing ratios of NOx and VOCs between LT
01:00–05:00 were not likely caused by chemical removal due to the rapid increase in
ozone. Regional transport of aged air masses (characterized by high ozone and low NOx
mixing ratios) may be responsible for the rapid decline in various VOC species in the
early morning of September 29. On the nighttime of September 29–30, the 450-m site
may be significantly impacted by surface fresh emissions as mixing ratios of ozone,
NOx, and VOCs all decreased between LT 19:00–01:00 and simultaneously increased
between LT 02:00–05:00. NOx and toluene mixing ratios generally increased between
LT 12:00–18:00 during September 27–29, which were quite different from their
average diurnal variation patterns during the whole campaign (Figures 3 and 4). As
discussed above, the 450-m site was located in the nocturnal residual layer during
September 27–29. Therefore, emissions of pollutants from surface sources could be
mixed upward to the measurement site only when the PBLH was higher than 450 m.
Furthermore, the PBL was relatively lower and rapidly shrank in the afternoon, leading
to the accumulation of chemical species at 450 m.

In summary, the VOCs measurements made by PTR-ToF-MS at the 450-m site

could be used to characterize variations in VOC species from their primary emissions
during the nighttime. Nevertheless, the oxidative degradation processes of VOCs in the
nighttime were not well captured. It is highly difficult to provide more information on
the nighttime chemistry of VOC species solely depending on their temporal variations.
We believe that the oxidative degradation of reactive VOC species did occur in the
nocturnal residual layer due to the coexistence of high concentrations of NOx and ozone.
Measurement techniques that targeting oxidation products (e.g., ToF-CIMS) and
numerical models should be jointly used to deeply analyze the nighttime chemistry of



VOCs in the nocturnal residual layer and quantitatively evaluate their impacts on
ambient air quality during the daytime.
## 4  Conclusions

Continuous measurements of VOCs mixing ratios were made by PTR-ToF-MS at

450 m on the CTT in PRD, China from August 18–November 5, 2020. In addition to
some specific VOC species (such as ethanol and monoterpenes) that were intensively
emitted by visitor-related sources, mixing ratios of most VOC species at 450 m were
generally lower than those at ground level. Due to intensive emissions from visitor-
related sources, mixing ratios of some VOC species were significantly higher on non-
working days than those on working days. The VOCs mixing ratios measured at 450 m
also exhibited different diurnal variations from those at ground level, indicating that
they were contributed by more mixed sources at larger spatial scales. Five sources,
namely daytime-mixed, visitor-related, vehicular+industrial, regional transport, and
VCP-dominated, were determined by the PMF model, contributing to 22%, 30%, 28%,
10%, and 11% of the average TVOC mixing ratio, respectively. In addition to the
daytime-mixed and visitor-related sources, the other three sources all had relatively
lower contributions on non-working days than on working days. The VCP-dominated
source contributed an average of 5.7 ppb to TVOC mixing ratios, which was
comparable to those reported in American cities (Gkatzelis et al., 2021). However, the
VCP-dominated source accounted for a much smaller fraction (11%) of measured
TVOC mixing ratios in this study than in U.S. cities (>50%). Therefore, the reduction
in anthropogenic VOC emissions from traffic and industrial sources are still priorities
of current air pollution control for Chinese cities. However, though smaller fraction of
VOCs contributed by VCPs was observed in this study compared to cities in U.S.
(McDonald et al., 2018; Gkatzelis et al., 2021), large fractions of key VOC species
(such as monoterpenes and some aromatic species) were attributed to the VCP-
dominated source. This may be important for formulating control strategies for specific



chemical species or when they are used as key tracers of certain emission sources.
The vertical distribution patterns of NOx, ozone, Ox, and PM$_{2.5}$ concentrations
were investigated using measurements made at four different heights on the CTT.
Vertical profiles of NOx and PM$_{2.5}$ generally exhibited negative gradients, while
vertical profiles of ozone generally demonstrated positive gradients. In addition, the
vertical gradients of air pollutants concentrations were larger in the nighttime than in
the daytime, predominantly owing to stronger stability of the nocturnal boundary layer.
The 450-m site was rarely located in the nocturnal residual layer as cloudy and rainy
weather dominated during the campaign. The selected case indicated that the NO$_3$- or
O$_3$-initiated degradation chemistry may be not the sole path for the removal of
unsaturated VOC species in the nocturnal residual layer. The degradation chemistry of
reactive VOC species in the nocturnal residual layer and their impacts on ground-level
air quality could be further investigated in combination with model simulations in
future studies.

## Data availability

The observational data used in this study are available from corresponding authors upon
request.

## Author contributions

XBL and BY designed the research. XBL, BY, SHW, CLW, JL, ZJL, XJH, YBHF, CLP,
CP, JPQ, CHW, YCY, MC, HDZ, WDY, XMW, and MS contributed to the data
collection and data analysis. XBL and BY performed the PMF analysis with
contributions from YXS, SXY, and SH. XBL and BY wrote the paper. All the coauthors
discussed the results and reviewed the paper.

## Competing interests

The authors declare that they have no conflict of interest.



## Acknowledgments

This work was financially supported by the National Key R&D Plan of China (grant No. 2019YFE0106300), Guangdong Natural Science Funds for Distinguished Young Scholar (grant No. 2018B030306037), the National Natural Science Foundation of China (grant No. 41877302, 42121004), Key-Area Research and Development Program of Guangdong Province (grant No. 2020B1111360003), China Postdoctoral Science Foundation (grant No. 2019M663367), Guangdong Innovative and Entrepreneurial Research Team Program (grant No. 2016ZT06N263), and Special Fund Project for Science and Technology Innovation Strategy of Guangdong Province (grant No.2019B121205004).



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



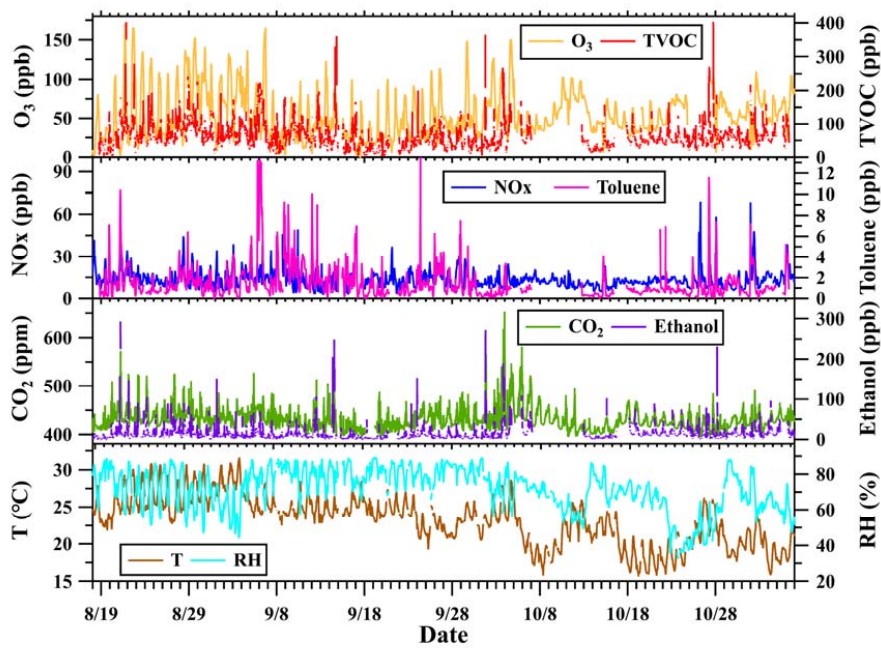

**Figure 1.** Time series of concentrations of some typical chemical species along with meteorological parameters during the CTT campaign. Temperature (T), relative humidity (RH), concentrations of ozone and NOx were measured at 488 m. Concentrations of VOCs, ethanol, and $CO_2$ were measured at 450 m.


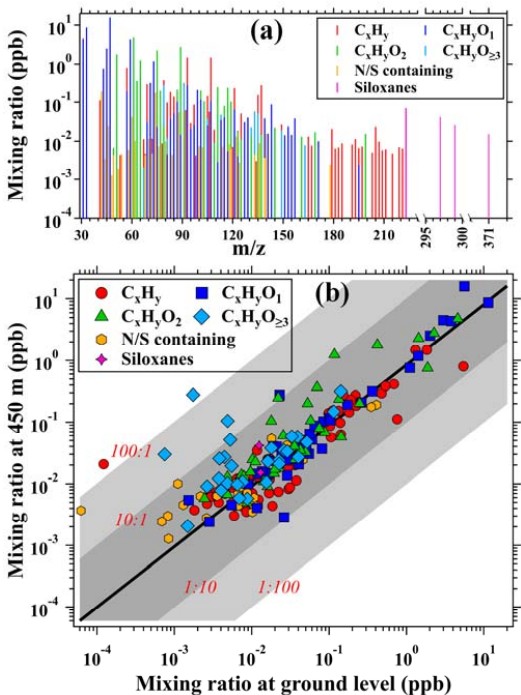

**Figure 2.** (a) Average mass spectra of VOCs (including 225 species) obtained by PTR-
ToF-MS during the CTT campaign. (b) Scatter plots of the average VOC mixing ratios
at 450 m during the CTT campaign versus those measured at the ground level during
the GIG campaign; The black solid line indicates the ratio of 1:1; The dark grey shaded
areas indicate the ratios of 10:1 and 1:10; The light grey shaded areas indicate the ratios
of 100:1 and 1:100. $C_xH_y$ refers to hydrocarbons. $C_xH_yO_1$ refers to VOC species
containing one oxygen atom. $C_xH_yO_2$ refers to VOC species containing two oxygen
atoms. $C_xH_yO_{\geq3}$ refers to VOC species containing more than three oxygen atoms. N/S
containing refers to VOC species containing nitrogen or sulfur atoms.

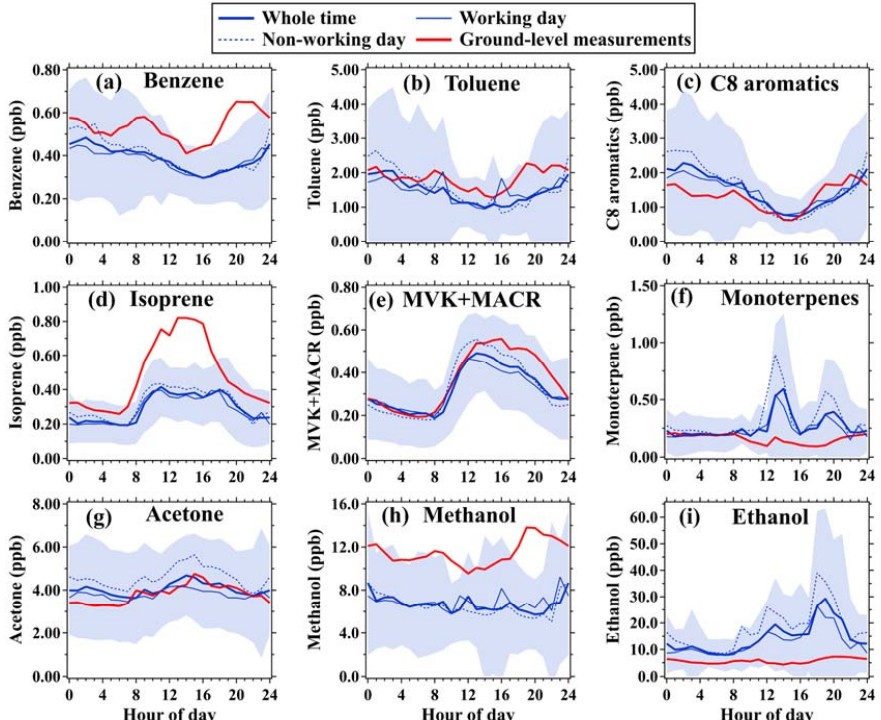

**Figure 3.** Diurnal variations in mixing ratios of selected VOC species measured by PTR-ToF-MS. Thick blue solid lines and shaded areas represent averages and standard deviations, respectively, during the CTT campaign (August 18–November 05, 2020). Red solid lines represent averages during the GIG campaign (September 11–November 19, 2018). Thin blue solid and dashed lines represent averages in working days and non-working (including weekends and public holidays) days, respectively, during the CTT campaign.

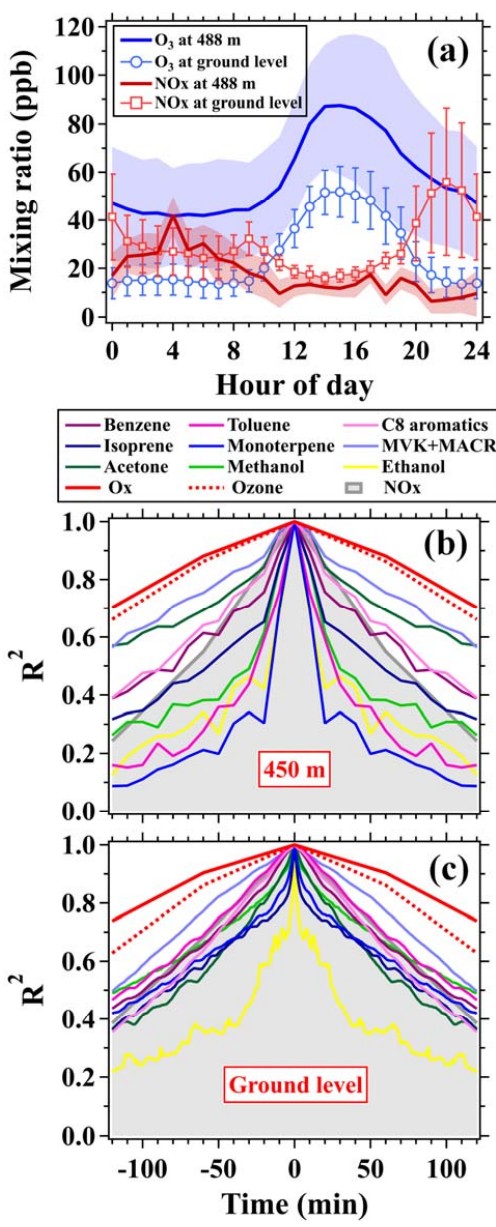

**Figure 4.** (a) Diurnal profiles of ozone and NOx mixing ratios measured at the 488-m

site (mean ± standard deviation) and the surface site (mean ± 0.5 standard deviation)

on the CTT. (b) Autocorrelation of the time series of ozone (488 m), NOx (488 m), Ox

(488 m), and selected VOC species (450 m) during the CTT campaign. (c)

Autocorrelation of the time series of the selected VOC species at the ground level



during the GIG campaign; Autocorrelation of the time series of ozone, NOx, and Ox in

panel (c) are calculated using the measurements made at the surface site of Canton

Tower during the CTT campaign.



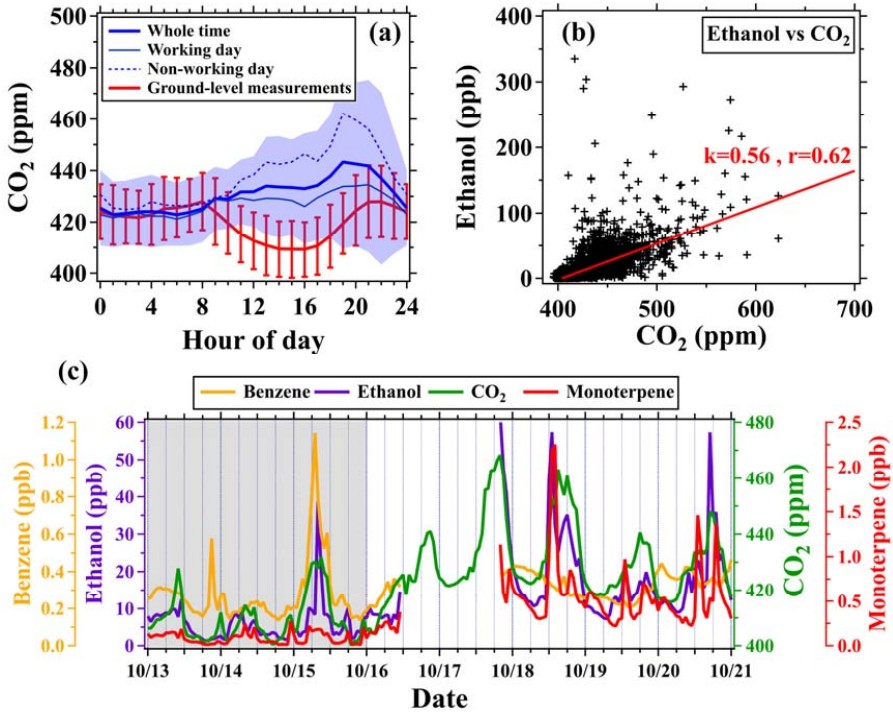

**Figure 5.** (a) Diurnal variations in $CO_2$ mixing ratios at 450 m and the ground level, respectively. (b) Scatterplots of ethanol versus $CO_2$ mixing ratios measured at 450 m during the CTT campaign; The ground-level $CO_2$ measurements were made in the GIG campaign. (c) Time series of benzene, ethanol, $CO_2$, and monoterpene mixing ratios measured at 450 m from October 13 to 21; The grey shaded area indicates the period (October 13–21) when the 450-m platform was closed due to the influence of Typhoon Kompasu.

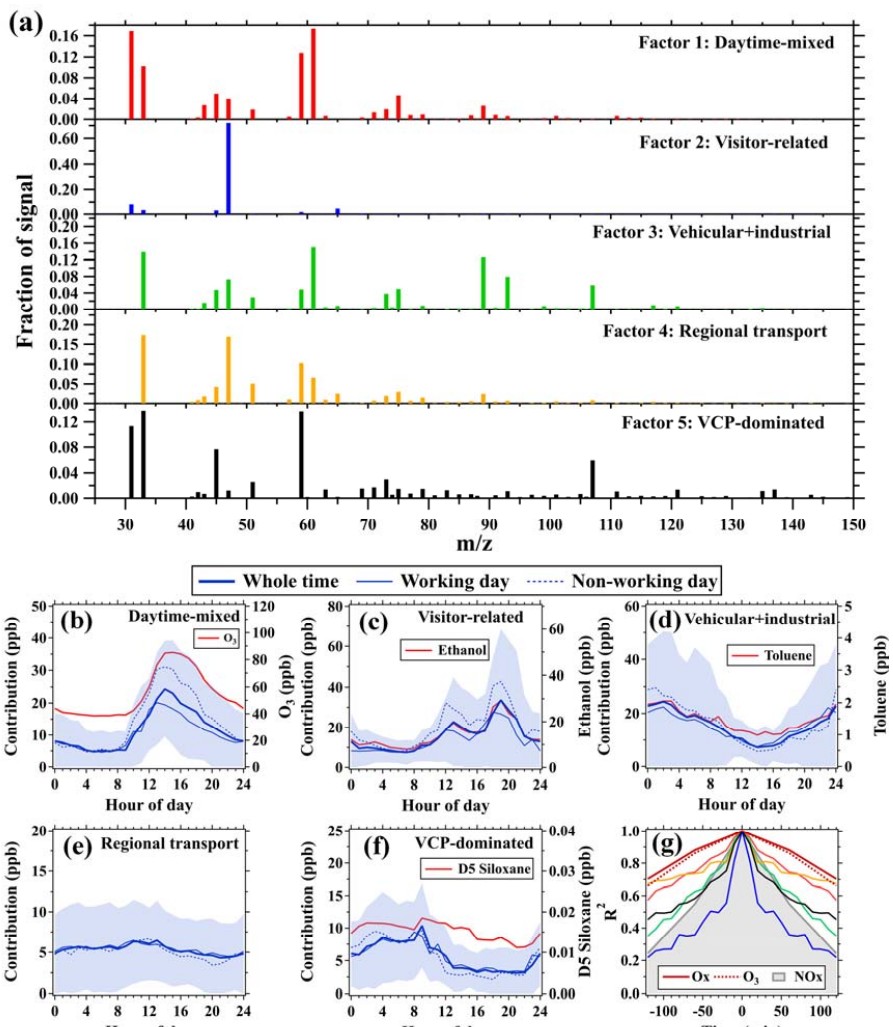

**Figure 6.** (a) Factor profiles (m/z $\leq$ 150) of the five PMF factors; Factor profiles with a full range of the mass spectra are provided in Figure S5. (b-f) Average diurnal profiles of the five PMF factors and source tracers. (g) Autocorrelation of the time series of the five PMF factors along with Ox, ozone, and NOx mixing ratios at 488 m.



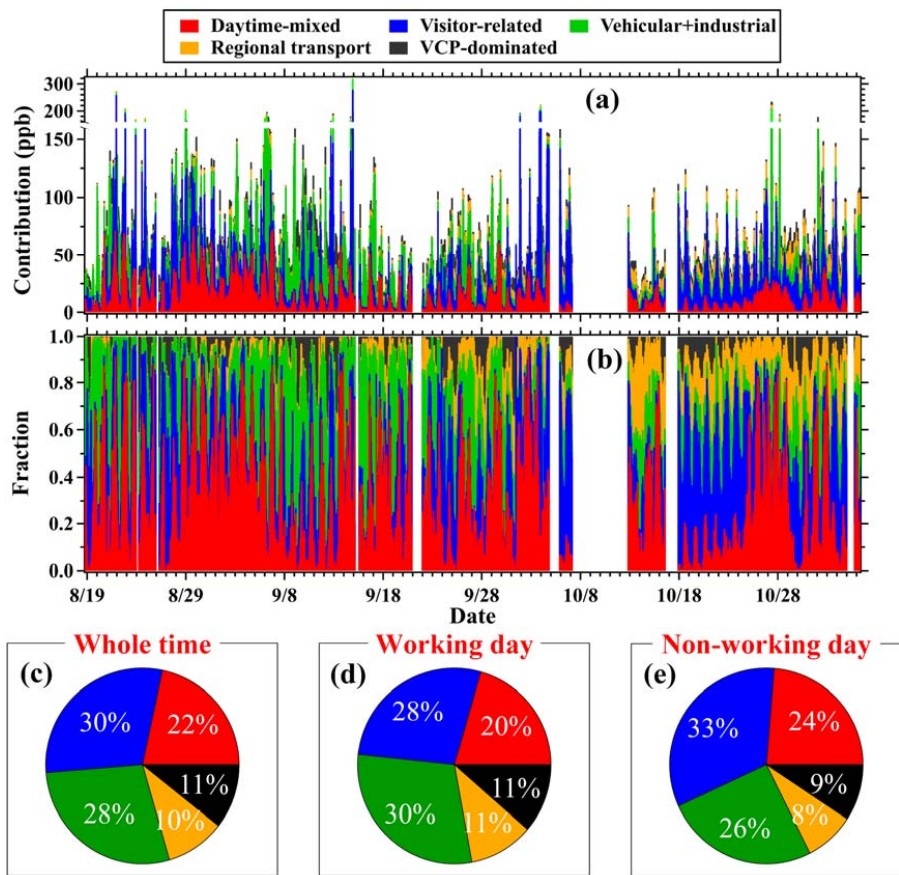

1141

**Figure 7.** (a-b) Stacked time series of factor fractions and factor contributions for the

PMF analysis; (c-e) Average contributions of the five PMF factors in the whole time,

working days, and non-working days.

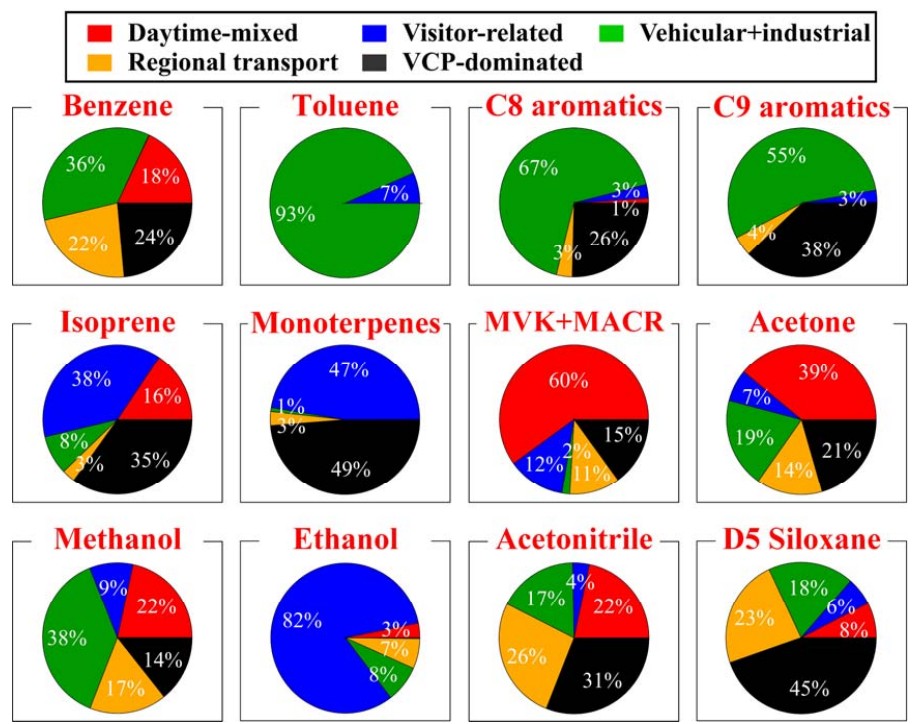

**Figure 8.** (a) Average contributions of the five PMF factors to the 9 selected VOC species during the CTT campaign.

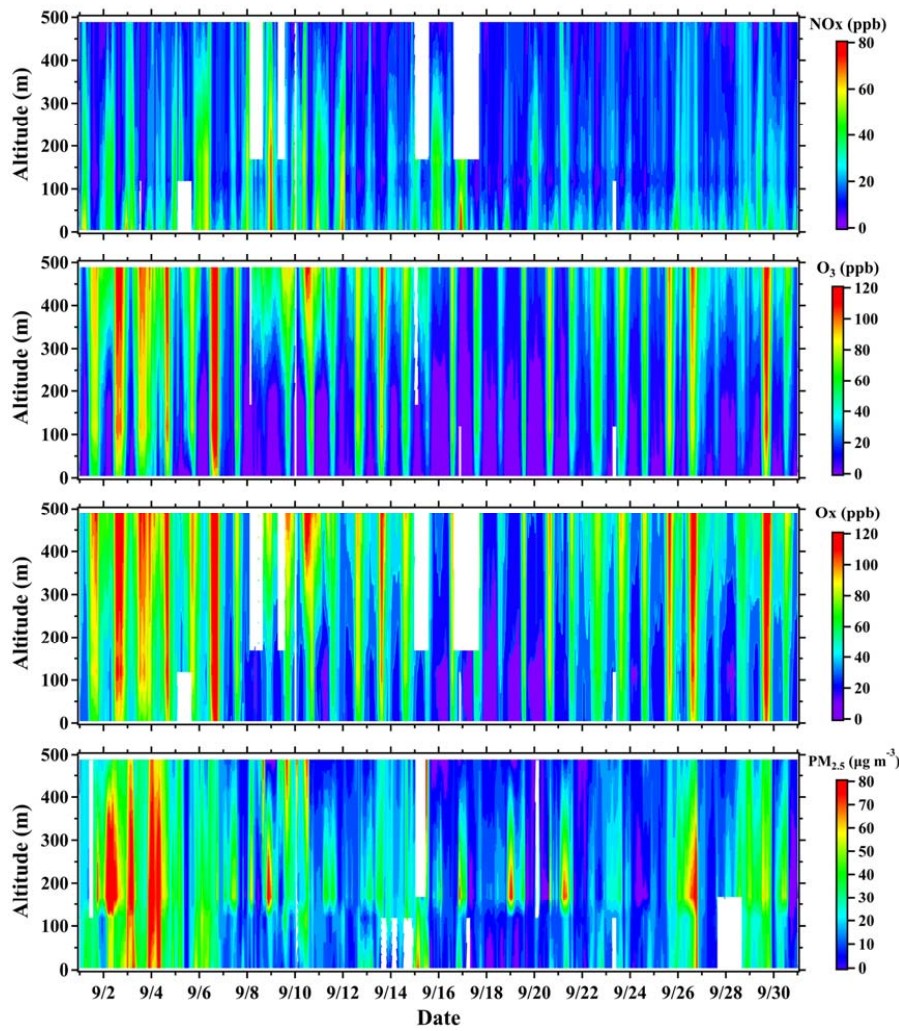

1148

**Figure 9.** Time series of vertical profiles for $O_3$, NOx, Ox ($O_3$+$NO_2$), and $PM_{2.5}$

concentrations in September during the CTT campaign. The contour plots are made

using the measurements from the four CTT sites (5 m, 118 m, 168 m, and 488 m).

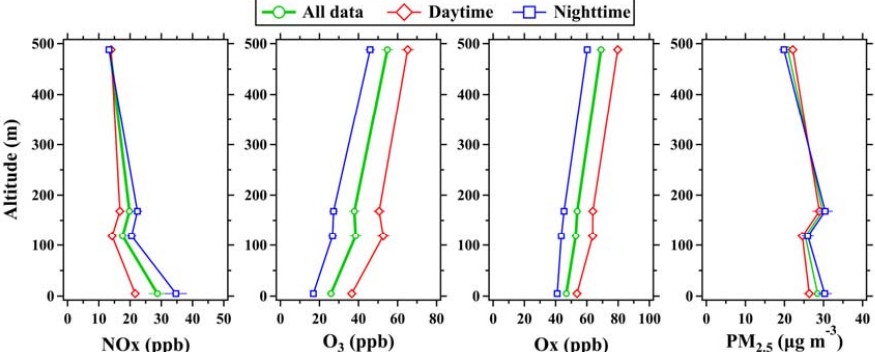

**Figure 10.** Average vertical profiles of $O_3$, NOx, Ox ($O_3$+$NO_2$), and $PM_{2.5}$ concentrations (mean ± 0.1 standard deviations) measured at the four CTT sites (5 m, 118 m, 168 m, 488 m) during the campaign. Daytime refers to the time between LT 08:00–18:00; nighttime refers to the time between LT 19:00–05:00.



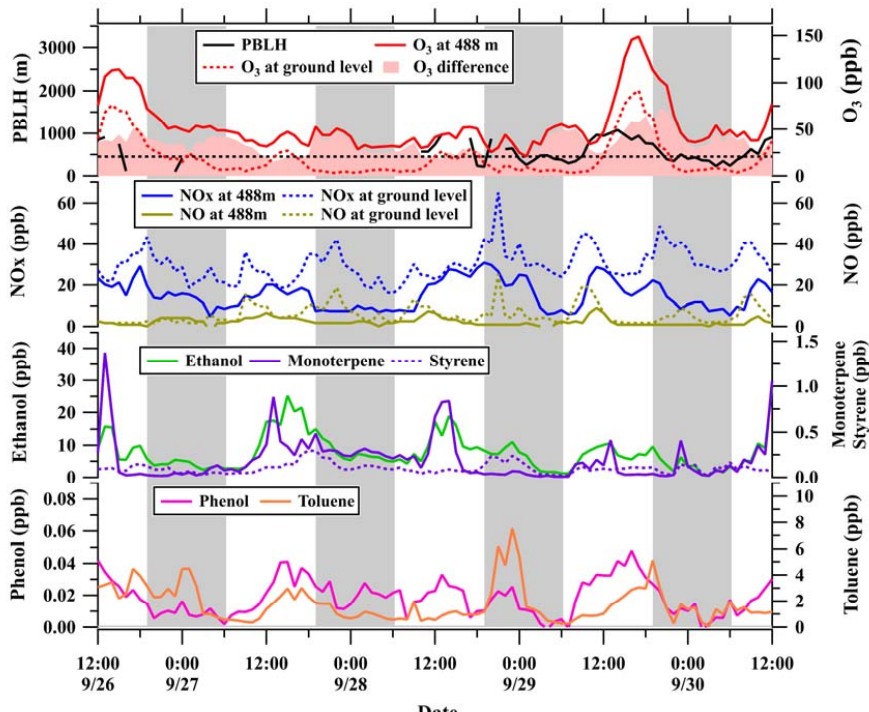

**Figure 11.** Time series of $O_3$, NOx, NO, ethanol, monoterpene, styrene, phenol, and toluene mixing ratios along with planetary boundary layer height (PBLH) during September 26–30. $O_3$ difference refers to the differences in ozone mixing ratios between 488 m and 5 m. Grey shaded areas indicate nighttime periods (LT 19:00–05:00).