# Peer review of "Variations and sources of volatile organic compounds (VOCs)"

_Atmospheric Chemistry and Physics, 2022_

## Author Comment (AC1)

* * *
**Response to Reviewer #1**
* * *
Li. et al. presents observations of volatile organic compounds (VOCs) collected from ~450m above ground level in Guangzhou, China, from mid-August to early-November. The authors were able to identify ~225 species using the Vocus PTR-MS. There were other measurements on the tower, including meteorology, CO2, O3, NO, NO2, NOx, and PM2.5; as well as, measurements of VOCs at a ground site ~5.7 km away from the tower and ceilometer for boundary layer height ~13.5 km from the tower. The authors look at both diurnal patterns and positive matrix factorization to identify sources of the VOCs, which include daytime-mixed, visitor-related, vehicular+industry, regional transport, and VCP-dominated. Also, the authors used autocorrelation to further identify the sources (which I found to be very informative). The authors use the PMF results to show the source contributions of the VOCs observed at the tower. Finally, they look at the profiled measurements of NOx, O3, and PM2.5 to look at the boundary layer dynamics throughout the measurement period and potential for residual layer vs nocturnal boundary layer chemistry.

As the authors noted, there are minimal measurements of VOCs at an elevated location (important for boundary layer dynamics and to look at regional background vs near-term sources), especially in China. That fact makes this paper of interest and importance for ACP. There are some aspects the authors could do to improve the paper, which I provide below, prior to publishing the paper in ACP.

**Reply:** We appreciate the reviewer for the valuable comments and suggestions, which are important for the improvement of our manuscript.

1) Further clarification in methods, specifically in the sampling.

1a) The authors noted that there was ~5 m long and was "extended to the outside wall of the observation room." However, it is not clear if this line is sampling inside the

tower or outside the tower. This aspect is important for clarification in other points presented below.

**Reply:** We appreciate the reviewer for the valuable suggestions. The ~5 m long tubing was connected to a reserved sampling port on the wall of the observation room to draw air sample from outside the tower. We have provided a picture (Figure S1(f)) to show the sampling port on the outside wall of the room and rephrased related sentences to make this point clearer. [see P: 7; L: 150-156]

*"To measure VOCs concentrations outside the tower, a ~5 m long Perfluoroalkoxy (PFA) Teflon tubing (OD: 1/4") was used to connect the inlet of the instrument and the sampling port (Figure S1). The PFA Teflon tubing has been proven to be effective in measuring ambient concentrations of VOCs (Deming et al., 2019; Liu et al., 2019) and has been widely used in field studies (de Gouw et al., 2003a; Hu et al., 2011; Wu et al., 2020a). Air sample in the tubing was drawn by a pump at a flow rate of ~5 L min$^{-1}$."*

1b) Was the sampling line heated or tested for potential losses of lower volatility / oxygenated species? There are some high molecular weight and/or oxygenated species that may have experienced lost.

**Reply:** We appreciate the reviewer for the valuable suggestions. The sampling line was not heated during the campaign. PFA Teflon tubing has been tested for potential losses of various VOC species in the literature (Deming et al., 2019; Liu et al., 2019) and has been proven to be the most suitable material so far to sample ambient air for VOCs analysis. In addition, the sampling flow rate was ~5 L min$^{-1}$, leading to a relatively small residence time (<3 s) of air sample in the tubing. PFA Teflon tubing has been widely used in the literature for VOCs analysis and thus was not tested in this study.

*Deming, B. L., Pagonis, D., Liu, X., Day, D. A., Talukdar, R., Krechmer, J. E., de Gouw, J. A., Jimenez, J. L., and Ziemann, P. J.: Measurements of delays of gas-phase*

*compounds in a wide variety of tubing materials due to gas–wall interactions, Atmos. Meas. Tech., 12, 3453-3461, 10.5194/amt-12-3453-2019, 2019.*

*Liu, X., Deming, B., Pagonis, D., Day, D. A., Palm, B. B., Talukdar, R., Roberts, J. M., Veres, P. R., Krechmer, J. E., Thornton, J. A., de Gouw, J. A., Ziemann, P. J., and Jimenez, J. L.: Effects of gas–wall interactions on measurements of semivolatile compounds and small polar molecules, Atmos. Meas. Tech., 12, 3137-3149, 10.5194/amt-12-3137-2019, 2019.*

1c) Further description of the observation level would be beneficial to understand the results presented. E.g., are the windows open or is the observation level "leaky"? If the line is sticking outside the tower (1a, unclear), it is surprising to see the "visitor-contribution" (more on that below). Thus, description about the observation level would be beneficial.

**Reply:** We appreciate the reviewer for the valuable suggestions. We have provided more descriptions and pictures (Figure S1) to introduce the observation level of the campaign on the CTT. The observation room is located below a ramp with stairs, namely the 450-m Look Out platform, on which visitors can walk around outside for a panorama of downtown Guangzhou but were not allowed to enter the observation room. The observation room has a reserved sampling port, as shown in Figure S1(f), so that the instruments can measure concentrations of chemical species outside the tower. As shown in Figure S1(g), a louver is located ~3 m below the sampling port. [see P: 6; L: 126-135] Detailed revisions are listed as follows:

*"The observation was conducted in a room (Figure S1) at the 450-m Look Out platform (Jin et al., 2022), which is a ramp with stairs and is located on the top of the main body of the CTT. The observation room is located below the ramp and a sampling port is reserved on the wall outside the tower. A louver is located ~3 m below the sampling port. The 450-m Look Out platform is a famous tourist attraction with an opening time of local time (LT, UTC+8) 10:00–22:30, and visitors could walk around for a panorama of downtown Guangzhou. On each day, there are two busiest tourist hours, roughly at LT 11:00–14:00 and 18:00–21:00, on the 450-m platform. In*

*addition, there are three restaurants between 376 and 423 m. The VOCs*
*measurements were interrupted during October 8–12 due to instrument malfunction.*"

1d) How was the data smoothed or extrapolated for the 3-D vertical profiles in Fig. 9? A description of that in Sect. 2.3 would be beneficial.

**Reply:** We appreciate the reviewer for the valuable suggestions. The observations made by different tower platforms were extrapolated for the 3-D vertical profiles using the bilinear method. That is, linear interpolations for concentrations of these pollutants were performed on both spatial (altitude) and temporal scales. Related descriptions have been provided in the manuscript. [see P: 8; L: 177-180]

"*Contour plots of vertical profiles of $NO_x$, ozone, $Ox$ ($O_3+NO_2$), and $PM_{2.5}$ concentrations were made using the bilinear method in Igor software (v8.04). Linear interpolations for concentrations of these pollutants were performed on both spatial (altitude) and temporal scales.*"

2) Something either in Sect. 3.1 or later that would be beneficial would be the OH reactivity (and maybe even $NO_3$-reactivity as some of this is later discussed in the context of residual layer chemistry) contribution of the compounds observed. As the authors note in line 606 - 609, though VCPs contributed a smaller amount of mixing ratio, some high reactivity compounds were in that class. As the combination of mixing ratio and reaction rate dictates the importance of the compound to urban chemistry, seeing how the classes weight out in reactivity space would be of great benefit.

**Reply:** We appreciate the reviewer for the valuable suggestions. We have calculated contributions of VOC categories (Figure 2) and PMF factors (Figure 7) to the total OH reactivity and provided related results and discussions in the revised manuscript. Contributions of different VOC categories and PMF factors to $NO_3$ reactivity are not discussed in this study because only a small fraction of VOC species can be effectively oxidized by $NO_3$ radicals.

In section 3.1, we calculated the average contributions of each VOC category to both total concentrations and OH reactivity. [see P: 10-11; L: 237-257] At 450 m, $C_xH_yO_1$ had the largest contribution (67%) to TVOC concentrations but only contributed to 40% of the total OH reactivity. $C_xH_y$ only accounted for 9% of TVOC concentrations but contributed to 37% of the total OH reactivity. At ground level, each VOCs category accounted for comparable fractions in total VOCs concentrations and OH reactivity to those measured at 450 m.

In section 3.4, we also discussed contributions of different PMF factors to the total OH reactivity. [see P: 16; L: 411-413] We find that the VCP-dominated (22%) and vehicular+industrial (23%) sources had comparable contributions to the total OH reactivity, even though large differences existed in their contributions to the TVOC concentrations (28% vs. 11%). These results further confirm that VCPs emissions should be given more attentions when making control strategies for VOCs in urban region.

[Figure]

*Figure 2 in the revised manuscript*

[Figure]

*Figure 7 in the revised manuscript*

3) Currently, Fig. 2 is too busy to interpret well and follow along with the authors' argument. What would be more informative would be to highlight the compounds that show XX difference between ground and elevated platform measurements (e.g., 50%, factor 2, or something else). Though seeing "family" (which is different than PMF) is informative, know which compounds are different can be equally important.

**Reply:** We appreciate the reviewer for the valuable suggestions. In this study, the VOCs measurements contain more than 200 species and thus it is quite difficult to discuss the differences in VOCs measurements between 450 m and the ground level at species levels. In section 3.2, we have presented and discussed the differences in concentrations and diurnal patterns of some typical VOC species. In addition, ratios of concentrations of 225 species measured at 450 m to those at ground level are provided

as a function of m/z in Figure S2. In the revised manuscript, the VOCs measurements made at the GIG site were mainly used to identify differences in contributions of different VOCs categories to the total concentrations and OH reactivity between 450 m and the ground level. These results can primarily reveal the differences in contribution sources of VOCs measurements between urban upper air and the ground level. However, the differences in concentrations of various VOCs between 450 m and the ground level were not highlighted because the two field campaigns were conducted at different times and sites.

[Figure]

*Figure S2(a) in the revised manuscript*

4) Fig. 3 and potentially other diurnal plots. Sometimes it's difficult to discern the differences the authors mention in the diurnal profiles either between ground-level and tower or working vs non-working. I would strongly recommend either as a figure in the main paper or a supplemental figure showing the ratios of these compounds and if they are statistically different or not.

**Reply:** We appreciate the reviewer for the valuable suggestions. In the revised manuscript, we have provided figures in SI showing diurnal variations in ratios of concentrations of the selected VOC species (Figure S3) and contributions of the five factors (Figure S8) between non-working and working days. The diurnal variations in ratios of concentrations of the selected VOC species between 450 m and the ground level were not highlighted because the two VOCs datasets were obtained at different times and sites.

As also suggested by the reviewer, the statistical significance levels ($p$ values) of differences were determined using the Student's t-test. [see P: 8: L:188-189]

5) Source analysis of the VOCs. It is surprising that authors are seeing such a large source of ethanol and $CO_2$ from visitors (comment 1c). It would be of use to better understand why this source is so large (is it due to experimental set up); whereas, e.g., they do not observe much VOCs from cooking when there are restaurants in the lower levels of the tower. Further, though the visitor profile is different from the VCP-dominated source, I would still recommend the authors be a little more cautious in this discussion. E.g., what the authors call the visitor-related compounds are also compounds that can generally be "VCP" in nature. Though it is a local source, it contributes/impacts the VCPs mixing ratio and chemistry.

**Reply:** We appreciate the reviewer for the valuable comments and suggestions. During the CTT campaign, the observation room was located on the 450-m Look Out platform, which is a ramp with stairs and is located on the top of the main body of the CTT. The observation room is located below the ramp, namely the 450-m Look Out platform, on which visitors can walk around outside for a panorama of downtown Guangzhou. Therefore, measured concentrations of VOCs were impacted by visitor-related emissions. We have provided more pictures (Figure S1) and descriptions to introduce the sampling site and the field campaign. The revisions about site description are listed as follows: [see P: 6: L:126-134]

*"The observation was conducted in a room (Figure S1) at the 450-m Look Out platform (Jin et al., 2022), which is a ramp with stairs and is located on the top of the main body of the CTT. The observation room is located below the ramp and a sampling port is reserved on the wall outside the tower. A louver is located ~3 m below the sampling port. The 450-m Look Out platform is a famous tourist attraction with an opening time of local time (LT, UTC+8) 10:00–22:30, and visitors could walk around for a panorama of downtown Guangzhou. On each day, there are two busiest tourist hours, roughly at LT 11:00–14:00 and 18:00–21:00, on the 450-m platform. In addition, there are three restaurants between 376 and 423 m."*

In addition, the CTT campaign was conducted in August-November of 2020, during which visitors must wear masks when visiting the CTT and ethanol-containing products (e.g., medicinal alcohol spray and 75%-ethanol bacteriostatic gel) were extensively used to prevent the spread of the COVID-19 pandemic. The total usage of ethanol-containing products was closely associated with the number of visitors, which can be manifested by the similar diurnal patterns of ethanol concentrations and the number of visitors at the 450-m platform. A detailed discussion on impacts of visitor-related emissions on VOCs measurements has been provided in Section 3.3. [see P: 14-15: L:345-379]

In addition, we also agree with the reviewer's opinion that the VOCs measurements may be affected by cooking emissions because the restaurants are located ~30 m below the observation site. Emission intensities of VOCs (e.g., monoterpenes) from cooking-related sources were also closely associated with the number of visitors. We have provided related discussions in the revised manuscript. [see P: 14-15: L:354-359]

"*In addition, the restaurants are located ~30 m below the observation site and emission intensities of VOCs (e.g., monoterpenes) from cooking-related sources were also closely associated with the number of visitors. Therefore, the VOCs measurements made at the 450-m platform were inevitably affected by visitor-related emissions, such as human breath, cooking, and volatilization of ethanol-containing and personal care products (Veres et al., 2013).*"

We also agree with the reviewer's opinion that the visitor-related compounds are also compounds that can generally be "VCP" in nature, which has been discussed in SI as follows. [see P: 22: L:119-125 in SI] In this study, we aim to investigate sources of VOCs in urban upper air as the VOCs measurements at high altitudes can be used to better characterize urban emissions at large spatial scales. It is very important to exclude VOCs emissions from local sources. Therefore, we believe that it is more suitable to separate visitor-related emissions from VCPs.

*"It should be noted that large fractions of ethanol emitted from personal care products were generally attributed to the VCP source in previous studies (McDonald et al., 2018; Gkatzelis et al., 2021). This is correct when the observation site was not affected by intensive emissions from a known source such as visitors at the 450-m platform. The visitor-related source was resolved in PMF to separate contributions of VCPs from those emitted by visitors."*

6) I'm very suprised by the fractional contribution pie chart in Fig. 7 vs the time series fractional contribution. E.g., it appears that visitor-related is typically on order 5-10% with maybe the observations in October being greater than 20% while daytime-mixed is the largest contributor during most of the study. Not sure if it is howing the data is being weighed/average.

**Reply:** We appreciate the reviewer for the valuable suggestions. We have rechecked the calculations for the PMF analysis and confirmed that the results in Figure 7 are correct. Figure 7(a-b) shows stacked time series of fractions and contributions of the five PMF factors at high time resolutions of 10 min, producing an illusion that the daytime-mixed source had larger contributions than other factors. We have rearranged the stack order of the five factors in Figure 7(a-b) in the revised manuscript.

[Figure]

*Figure 7(a-b) in the revised manuscript*

Minor

1) The authors use the word "significantly" or "significant" throughout the text. I would strongly recommend the authors use a different word when there is a difference unless they have conducted statistical analysis (e.g., Student's T-Test) to determine if there is significant difference.

**Reply:** We appreciate the reviewer for the valuable suggestions. In the revised manuscript, the Student's t-test was used to determine the statistical significance levels of differences when performing comparisons. Otherwise, the words "significantly" and "significant" have been replaced with other words (e.g., "notably" and "markedly").

2) Another statistics aspect to be careful in includes when the authors look at the r values and state that it is well correlated. E.g., an r value of 0.62 (line 336) indicates that $CO_2$ can explain ~38% of the observed ethanol mixing ratios.

**Reply:** We appreciate the reviewer for the valuable comments. As suggested by the reviewer, we have provided significance levels ($p$ values) when performing comparison (t-test) or fitting (F-test) analysis in the revised manuscript. As for the correlation between $CO_2$ and ethanol, the $p$ value of the linear fitting was lower than 0.01 (p<0.01). In addition, the correlation coefficient ($r$) generally decreases with increased sample size. In this study, the correlation coefficient between mixing ratios of $CO_2$ and ethanol was calculated at 10-min time resolutions (Figure 5(b)) and the sample size was more than 10,000. Therefore, we think that the mixing ratios of $CO_2$ and ethanol was well correlated with the $r$ value of 0.62. [see P: 15: L:363-366]

3) Fig. 6, the PMF factors having similar colors to the ground and tower observations makes it difficult to interpret. I would recommend different colors for the PMF results.

**Reply:** We appreciate the reviewer for the valuable suggestions. We have changed colors of the VOCs categories for the ground and tower observations in Figure 2. The present colors for PMF analysis have a good contrasting effect and remain unchanged. [see P: 40]

---

## Author Comment (AC2)

* * *
**Response to Reviewer #2**
* * *
Li et al. report interesting time-resolved VOC measurements on the Canton Tower in Guangzhou, China using a QiTOF PTRMS. The Guangzhou region is impacted by air pollution and underrepresented in atmospheric observations which is why data from this region are a valuable opportunity to learn about the processes and regional sources. The identification of these sources from the data must be challenging due to chemical complexity. PMF was utilized to help with this task and five factors were obtained from the PMF source apportionment conducted on the 225 VOC ions which were assigned different mostly anthropogenic categories. The observations focus on abundant VOCs and correlations with inorganic pollutants which might be helpful to understand variability of pollutants, oxidation and sources in the atmosphere. The text reads generally clear and there are rich figures to illustrate observations and thoughtful analyses. However, I'd have some comments and suggestions before the paper is published, and in particular there seem to be some issues or inconsistencies in a few compounds and PMF interpretations which hopefully can easily be addressed.

**Reply:** We appreciate the reviewer for the valuable comments and suggestions to improve the quality of our manuscript.

**Major**

Ethanol (e.g. Fig 1) is rather high as for the outdoor atmosphere and what I am most surprised with is that it is attributed to human emissions from tower visitors. If ethanol was emitted by human visitors on the tower why were siloxanes, not co-emitted? It is difficult to get convinced that such a large concentration of ethanol comes from human breath as is suggested in the SI Page 19. Was maybe a sanitizer dispenser in the vicinity to the sampling inlet? A picture or at least a schematic of the inlet location would be useful (as well as the info if the particle filter was used or not).

**Reply:** We appreciate the reviewer for the valuable suggestions. As shown in Figure S1, the observation room is located on the 450-m Look Out platform below the ramp, on which visitors can walk around. Therefore, measured concentrations of VOCs were impacted by visitor-related emissions, such as human breath, cooking, and the volatilization of personal care products. In addition, the campaign was conducted in August-November of 2020, during which stringent control measures were implemented to prevent the spread of the COVID-19 pandemic. For example, visitors must wear masks and ethanol-containing products (e.g., medicinal alcohol spray and 75%-ethanol bacteriostatic gel) were extensively used to wipe public utilities. The total usage of ethanol-containing products had a strong dependence on the number of visitors, which can be manifested by the similar diurnal patterns of ethanol concentrations and the number of visitors at the 450-m platform. Therefore, we concluded that the enhanced ethanol concentrations were mainly attributed to visitor-related emissions, predominantly from the volatilization of ethanol-containing products and personal care products. The human breath, cooking, and volatilization of personal care products also contribute to measured ethanol concentrations but may not account for comparable fractions to that the visitor-related source. A detailed discussion on impacts of visitor-related emissions on VOCs measurements has been provided in Section 3.3. [see P: 14-15: L:345-379]

In the revised manuscript, we have provided more pictures (Figure S1) and descriptions to introduce the sampling site and the field campaign. [see P: 6: L:126-135]

"*The observation was conducted in a room (Figure S1) at the 450-m Look Out platform (Jin et al., 2022), which is a ramp with stairs and is located on the top of the main body of the CTT. The observation room is located below the ramp and a sampling port is reserved on the wall outside the tower. A louver is located ~3 m below the sampling port. The 450-m Look Out platform is a famous tourist attraction with an opening time of local time (LT, UTC+8) 10:00–22:30, and visitors could walk around for a panorama of downtown Guangzhou. On each day, there are two busiest*

*tourist hours, roughly at LT 11:00–14:00 and 18:00–21:00, on the 450-m platform. In addition, there are three restaurants between 376 and 423 m. The VOCs measurements were interrupted during October 8–12 due to instrument malfunction.*"

The concentration data at least for ethanol are inconsistent. Ethanol concentration is shown in multiple figures with overlapping times in Fig 1 and 11. Ethanol concentration is different in those figures (e.g. 9/30, 9/29).

**Reply:** We appreciate the reviewer for the valuable suggestions. Reasons for causing the differences in ethanol concentrations in Figures 1 and 11 in the original manuscript are the differences in time resolutions of ethanol concentrations. In the original manuscript, Figure 1 shows time series of ethanol concentrations at time resolutions of 10 min and Figure 11 shows hourly mean concentrations of ethanol. In the revised manuscript, hourly mean concentrations of ethanol, as well as other VOC species, were used in Figure 1 to keep consistent with those in Figure 11.

The lack of dependence on wind direction (L397) is surprising and makes me wonder if the instrument may have been sampling from indoors, indoor plumes, or if there may have been a leak in the line or other factor which would explain high ethanol, and acetic acid concentrations. Wind roses or polar plots for VOCs would also be useful to help in clarifying these issues and help in source interpretations.

**Reply:** We appreciate the reviewer for the valuable suggestions. Indeed, it would be very useful to show the dependence of VOCs concentrations on wind direction. However, wind speed and wind direction were not measured at 450 m during the campaign because the measurements of winds can be easily changed by the structure of the CTT. This is the reason why we performed a cluster analysis of backward trajectories of air masses in this study to roughly investigate the dependence of VOCs concentrations and factor contributions on wind direction. [see P: 26 in SI]

In addition to instrument maintenance (usually several hours), there were no people in the observation room. Therefore, concentrations of various VOC species will exhibit insignificant diurnal variations if there is a leak in the sampling tubing.

The PMF analysis results seem surprising to me. It almost seems like the same compounds and factor profiles appear in every factor (Figure S5). Methanol and acetone appear in multiple factors as most abundant. Visitor factor is lacking siloxanes. Given it is the QiTOF with very high sensitivity, I'd expect the clearer factors including biogenic, oxygenated biogenic, and cooking could be obtained without merging. I wonder if optimizing uncertainties or dividing the dataset into shorter periods could further help in those source interpretations.

**Reply:** We appreciate the reviewer for the valuable suggestions. As known, many OVOC species have very complex sources (e.g., various primary emission sources and secondary formation) in urban environments. Therefore, many OVOC species were abundant in different PMF factors and this is consistent with the results in previous studies (Gkatzelis et al., 2021; Pallavi et al., 2019) that used measurements of PTR-ToF-MS for PMF analysis. Therefore, likely sources of the PMF factors based on PTR-ToF-MS measurements were usually determined using the combination of factor profile and diurnal variation patterns of factor contributions. As for contributions of the visitor-related source, the human breath and volatilization of personal care products may only account for a small fraction. As a typical tracer of VCPs, D5 siloxane was mainly attributed to the VCP-dominated source, as shown in Figure 8.

As also suggested by the reviewer, we have tried performing the PMF calculation many times by changing the uncertainties of different VOCs species and the analyzing time scales and we believe that the present results were optimal according to our analysis.

*Gkatzelis, Georgios I., Coggon, Matthew M., McDonald, Brian C., Peischl, Jeff, Gilman, Jessica B., Aikin, Kenneth C., Robinson, Michael A., Canonaco, Francesco, Prevot, Andre S. H., Trainer, Michael, and Warneke, Carsten: Observations Confirm that Volatile Chemical Products Are a Major Source of Petrochemical Emissions in U.S. Cities, Environmental Science & Technology, https://doi.org/10.1021/acs.est.0c05471, 2021.*

*Pallavi, Sinha, B., and Sinha, V.: Source apportionment of volatile organic compounds in the northwest Indo-Gangetic Plain using a positive matrix factorization model, Atmospheric Chemistry and Physics, 19, 15467-15482, https://doi.org/10.5194/acp-19-15467-2019, 2019.*

Minor

I really like Table S1 with nicely tabulated masses and formulas identifications and sensitivities with clear note which ion was explicitly calibrated. I think it is exemplary for all PTRMS papers. I only have minor suggestions here: 69.03 should be furan (not fural), 51.99 monochloramine, 71.03 add MACR. Did you not see D5 fragment at 355.06? This is extremely surprising as is the unexpected D5 fragment at 299. I am again curious what the E/N ratio was.

**Reply:** We appreciate the reviewer for pointing out the mistakes and providing valuable suggestions. We have revised or added compounds for corresponding signals in Table S1. Ion signals at m/z 299.06 and 355.06 are all D5 fragments in PTR-ToF measurements, but measurements of m/z 355.06 were excluded from the PMF analysis due to its relatively low signals during the campaign.

As suggested by the reviewer, we have provided more information to introduce the PTR-ToF-MS used in our study. In $H_3O^+$ mode, the PTR-ToF-MS was operated with a drift tube pressure of 3.8 mbar, a drift tube temperature of 120 °C, and a drift tube voltage of 760 V, resulting in an E/N value of ~120 Td. [see P: 7: L:143-146]

It is mentioned that ethyl acetate was one of the most abundant VOCs, but how did the authors exclude butyric acid or subtracted it from the signal?

**Reply:** We appreciate the reviewer for the valuable comments. In addition to ethyl acetate, ion signals measured at m/z 89.06 were also likely contributed by butyric acid because PTR-ToF-MS cannot resolve signals of isomers. In this study, we attributed ion signals of m/z 89.06 to ethyl acetate for two important reasons. First, in addition to formic acid, molecules of other organic acids can easily fragment when measured by PTR-ToF-MS (Haase et al., 2012; Praplan et al., 2014). Second, ambient

concentrations of butyric acid were significantly lower than those of ethyl acetate in urban environments, as ethyl acetate is frequently used as solvents for various applications (Zheng et al., 2013). For example, the mean concentration of butyric acid measured by a I ToF-CIMS (I ToF-CIMS could accurately measure concentrations of butyric acid) in the GIG campaign was only $0.24\pm0.17$ ppb (Ye et al., 2021), which was much lower than that of ethyl acetate ($2.27\pm2.36$ ppb) measured by the PTR-ToF-MS.

*Haase, K. B., Keene, W. C., Pszenny, A. A. P., Mayne, H. R., Talbot, R. W., and Sive, B. C.: Calibration and intercomparison of acetic acid measurements using proton-transfer-reaction mass spectrometry (PTR-MS), Atmos. Meas. Tech., 5, 2739-2750, 10.5194/amt-5-2739-2012, 2012.*

*Praplan, A. P., Hegyi-Gaeggeler, K., Barmet, P., Pfaffenberger, L., Dommen, J., and Baltensperger, U.: Online measurements of water-soluble organic acids in the gas and aerosol phase from the photooxidation of 1,3,5-trimethylbenzene, Atmos. Chem. Phys., 14, 8665-8677, 10.5194/acp-14-8665-2014, 2014.*

*Ye, C., Yuan, B., Lin, Y., Wang, Z., Hu, W., Li, T., Chen, W., Wu, C., Wang, C., Huang, S., Qi, J., Wang, B., Wang, C., Song, W., Wang, X., Zheng, E., Krechmer, J. E., Ye, P., Zhang, Z., Wang, X., Worsnop, D. R., and Shao, M.: Chemical characterization of oxygenated organic compounds in the gas phase and particle phase using iodide CIMS with FIGAERO in urban air, Atmos. Chem. Phys., 21, 8455-8478, https://doi.org/10.5194/acp-21-8455-2021, 2021.*

*Zheng, J., Yu, Y., Mo, Z., Zhang, Z., Wang, X., Yin, S., Peng, K., Yang, Y., Feng, X., and Cai, H.: Industrial sector-based volatile organic compound (VOC) source profiles measured in manufacturing facilities in the Pearl River Delta, China, Sci Total Environ, 456-457, 127-136, https://doi.org/10.1016/j.scitotenv.2013.03.055, 2013.*

Nowhere in the text or SI PTRMS parameters are shown. It would be great to provide at least E/N, which is a common practice in a PTRMS papers, because it helps interpret the VOC data and estimate expected fragmentations.

**Reply:** We appreciate the reviewer for the valuable suggestions. The E/N of PTR-ToF-MS was ~120 Td during the campaign. In the original manuscript, these information were not provided because they have been introduced in our previous papers (Wang et al., 2020a; Wu et al., 2020a; He et al., 2022). We have provided related information in the revised manuscript. [see P: 7: L:143-150]

"*In $H_3O^+$ mode, the PTR-QiToF-MS was operated with a drift tube pressure of 3.8 mbar, a drift tube temperature of 120 °C, and a drift tube voltage of 760 V, resulting in an E/N (E refers to electric field and N refers to number density of buffer gas in the drift tube) value of ~120 Td (Townsend). Raw data of PTR-ToF-MS were processed and analyzed using Tofware software (Tofwerk AG, v3.0.3) and please refer to our previous works (Wang et al., 2020a; Wu et al., 2020a) for details*"

Fig. 3: I wonder why the traffic rush hours are not clearly seen on the plots presenting aromatics. The period was probably not during the lockdown if the visitors are allowed in the tower. Ethanol and monoterpenes coincide with lunch and dinner so might be cooking emissions(?).

**Reply:** We appreciate the reviewer for the valuable suggestions. As shown in Figure 3, diurnal profiles of aromatics (e.g., benzene and toluene) usually exhibited bimodal patterns in morning and evening rush hours at ground level due to the combined effects of enhanced traffic emissions and strong atmospheric stability. However, the measurements of VOCs at 450 m were less affected by traffic emissions from the ground in morning and evening rush hours due to enhanced atmospheric stability (largely limiting the vertical mixing of surface emissions).

The CTT campaign was conducted in August-November of 2020, during which ethanol-containing products (e.g., medicinal alcohol spray and 75%-ethanol bacteriostatic gel) were still extensively used to prevent the spread of the COVID-19

pandemic. For example, medicinal alcohol (75%) spray was widely used to wipe public utilities and 75%-ethanol bacteriostatic gel was extensively used as sanitizer for hands. [see P: 14: L:346-352] The total consumption of ethanol-containing products was dependent on the number of visitors on the CTT. In addition, we also agree with the reviewer's opinion that the VOCs measurements may be affected by cooking emissions because the restaurants are located ~30 m below the observation site. Emission intensities of VOCs (e.g., ethanol and monoterpenes) from cooking-related sources were also closely associated with the number of visitors. We have provided related discussions in the revised manuscript. [see P: 14-15: L:354-359]

Fig. 5: Super interesting effect of Typhoon on VOC concentration reduction. I wonder about the dependence of the observed concentrations on the wind speed.

   **Reply:** We appreciate the reviewer for the valuable suggestions. A cluster analysis of backward trajectories of air masses [see P: 26 in SI] was performed in this study to roughly investigate the dependence of VOCs concentrations and factor contributions on wind direction due to the lack of wind measurements during the campaign.

   In Figure 5, what we want to discuss was not the dilution or transport effect of the Typhoon on the measurements of VOCs but impacts of the closure of the 450-m platform on changes in measured concentrations of various VOC species. The 450-m platform was closed on October 13-15 due to the influence of Typhoon Kompasu. On these days, mixing ratios of ethanol, $CO_2$, and monoterpenes exhibited similar variation patterns to benzene (a typical tracer of traffic emissions). However, mixing ratios of ethanol, $CO_2$, and monoterpenes exhibited quite different variation patterns from benzene when the 450-m platform was re-open (October 16–21). These results further confirm the fact that visitor-related emissions contributed to the VOCs measurements when the 450-m platform was open.

 I could not find in the main text or SI how exactly the PTRTOF data were processed, what software was used for mass scale alignment, peak fitting, as well as quality

control, formula assignment, compound inference, total number of ions detected and steps taken for ion reduction or any abundance filters. This info would be greatly appreciated in the methods or SI.

**Reply:** We appreciate the reviewer for the valuable suggestions. Raw data of PTR-ToF-MS were processed (e.g., mass scale alignment and peak fitting) and analyzed (e.g., formula assignment, mass calibration, and compound inference) using Tofware software (Tofwerk AG, v3.0.3). Signals of 3035 ions with m/z up to 510 were obtained at time resolutions of 10 s. Original ion signals measured by PTR-ToF-MS may be affected by the change in ambient humidity. Therefore, effects of changes in humidity on measured signals of various VOCs were excluded using humidity-dependence curves determined in the laboratory. We have provided these descriptions in the revised manuscript. [see P: 7: L:146-150] More detailed information on the processing of PTR-ToF-MS data has been provided in our previous papers (Wu et al., 2020; Wang et al., 2020) and was thus not repeated in the present work.

"*Raw data of PTR-ToF-MS were processed and analyzed using Tofware software (Tofwerk AG) and please refer to our previous works (Wu et al., 2020a; Wang et al., 2020a) for details. Signals of 3035 ions with m/z up to 510 were obtained at a time resolution of 10 s.*"

*Wu, C., Wang, C., Wang, S., Wang, W., Yuan, B., Qi, J., Wang, B., Wang, H., Wang, C., Song, W., Wang, X., Hu, W., Lou, S., Ye, C., Peng, Y., Wang, Z., Huangfu, Y., Xie, Y., Zhu, M., Zheng, J., Wang, X., Jiang, B., Zhang, Z., and Shao, M.: Measurement report: Important contributions of oxygenated compounds to emissions and chemistry of volatile organic compounds in urban air, Atmos. Chem. Phys., 20, 14769-14785, https://doi.org/10.5194/acp-20-14769-2020, 2020.*

*Wang, C., Yuan, B., Wu, C., Wang, S., Qi, J., Wang, B., Wang, Z., Hu, W., Chen, W., Ye, C., Wang, W., Sun, Y., Wang, C., Huang, S., Song, W., Wang, X., Yang, S., Zhang, S., Xu, W., Ma, N., Zhang, Z., Jiang, B., Su, H., Cheng, Y., Wang, X., and Shao, M.: Measurements of higher alkanes using NO+ chemical ionization in PTR-ToF-MS:*

*important contributions of higher alkanes to secondary organic aerosols in China, Atmos. Chem. Phys., 20, 14123-14138, https://doi.org/10.5194/acp-20-14123-2020, 2020.*

I am quite intrigued by the chromium ion reported in the SI Table. Did its signal show some ambient structure or was it coming from the ion source? There are many cool molecules in this table that could be useful to explore as source markers and complement PMF.

**Reply:** We appreciate the reviewer for the valuable suggestions. Figure R1 shows the time series of mixing ratios of m/z 84.9 ($CrO_2$) from October 24 to November 3, 2020, during the CTT campaign. Mixing ratios of m/z 84.9 exhibited similar, to some extent, variations to those of ambient humidity. Therefore, we agree with the reviewer's opinion that measurements of ion signals at m/z 84.9 were possible impurities contributed by the ion source. However, the mean concentration of the signal at m/z 84.9 was only 7 ppt during the campaign, which had negligible impacts on the results of this study and thus was kept in the analysis.

In this work, online measurements of more than 200 VOC species made at the 450 m site were predominantly attributed to five sources using the PMF analysis method. We admit that the PMF method has shortcomings in identifying sources of VOCs and each of the five PMF factors may contain contributions from various emission sources. For example, the daytime-mixed source may contain contributions from biogenic emissions, secondary formation, evaporation of solvents, and other sources that have strong dependences on solar radiation, temperature, and humidity. To clearly identify contribution sources of VOCs measurements at species levels in urban upper air, VOCs measurements with more species (e.g., NMHCs, organic acids, and higher alkanes) are required. This is also one of the primary works in our upcoming studies.

Technical

L 75 "were" should be "are"'

**Reply:** We appreciate the reviewer for pointing out this mistake and it has been corrected. [see P: 4: L:76]

L109 "by far" should be "so far"

**Reply:** We appreciate the reviewer for pointing out this mistake and it has been corrected. [see P: 5: L:110]

---

## Author Response (AR2)

**Response to Reviewer #1**

Li et al. present observations of VOCs measured on top of a 450 m tower. As stated in prior review, these measurements are of value to better understand how VOCs behave at higher levels in the boundary layer, as there have been minimal constraints.

The authors have done a great job in addressing all the comments from both reviewers. The new analysis and figures have greatly improved the overall manuscript and makes it of value to the ACP community.

**Reply:** We appreciate the reviewer for the valuable comments.

However, one minor aspect should be addressed with the new information provided by the authors. Looking at their S1 figure, which is greatly appreciated and useful in understanding the set-up, there is concern about the location of the inlet. From the figure, it looks like there is a large amount of roofing between the inlet and the edge of the building. Thus, two potential aspects that maybe biasing the results I believe should be mentioned:

(1) How much could off-gassing from the roof impact the measurements?

**Reply:** We appreciate the reviewer for the valuable comments and suggestions. In the last version of the SI file, some of the bottom parts of Figure S1(f) were covered to show the window of the observation room (Figure S1(g)), resulting in an illusion that there is a large amount of roofing between the inlet and the edge of the building. We have provided a clearer picture in Figure S1 to show the location of the sampling port in the revised SI file. Actually, the "roofing" is the orbit of the Bubble Tram on the 450 m platform (Figures R1 and S1(d)). The orbit of the Bubble Tram is located on the open hollow steel structure of the Canton Tower and is ~10 m higher when passing over the sampling inlet. Therefore, the off-gassing from the Bubble Tram could be largely attributed to emissions from visitor-related sources and had

weaker impacts on the VOCs measurements in comparison to those from visitors on the 450 m platform.

[Figure]

**Figure R1.** Picture showing the orbit of the Bubble Tram on the 450 m Look Out platform. The picture was obtained from the official website of the Canton Tower: https://www.cantontower.com/en/sightseeing/ferriswheel/.

(2) How could local eddies being created by that roof and thus potentially sampling more local VOCs than urban background impact the measurements?

  **Reply:** We appreciate the reviewer for the valuable comments and suggestions. As shown in Figures R1 and S1(d), the orbit of the Bubble Tram is quite narrow (~1 m) and is built on the open hollow steel structure of the Canton Tower. Under these circumstances, the Bubble Tram is unlikely to create eddies significantly affecting the VOCs measurements at the sampling port.

[Figure]

*Figure S1. in the revised SI file.*

Another minor comment from what the authors included in their comments. I like how the authors addressed the concerns about the visitors PMF in that it is a local source and would not be typical of the upper boundary layer measurements. It may be of value, in text, to state that and when comparing concentrations and reactivity, to have with and without the visitors PMF to better reflect how this factor may not be important for typical upper boundary layer VOCs.

**Reply:** We appreciate the reviewer for the valuable comments and suggestions. As suggested by the reviewer, we have provided discussions in the revised manuscript to highlight the local characteristics of the visitor-related source in PMF analysis. [see P: 16; L: 396-398] We agree with the reviewer's opinion that contributions of the visitor-related source should be excluded when highlighting contributions of other typical sources to the VOCs measurements or sources of typical VOC species in the

upper boundary layer. Therefore, contributions of the visitor-related source were considered when discussing contributions of the VCP-dominated source in TVOC mixing ratios [see P: 16; L: 405-411] and sources of the ethanol measurements [see P: 19; L: 478-489] in the manuscript. Contributions of the visitor-related source were not excluded in most discussions of the paper as we believe that it is also a significant contributor to ambient VOCs concentrations at ground level (particularly during the outbreak of the COVID-19 pandemic) and should be given more concern in future studies.

"*It should be noted that visitor-related emissions belonged to highly local sources on the 450 m platform and were not typical of the VOCs measurements in the upper boundary layer.*"